# Structural basis for bacterial energy extraction from atmospheric hydrogen

Rhys Grinter[1,15✉], Ashleigh Kropp[1,15], Hari Venugopal[2], Moritz Senger[3], Jack Badley[4], Princess R. Cabotaje[3], Ruyu Jia[4], Zehui Duan[5], Ping Huang[3], Sven T. Stripp[6], Christopher K. Barlow[7,8], Matthew Belousoff[9], Hannah S. Shafaat[10], Gregory M. Cook[11], Ralf B. Schittenhelm[7,8], Kylie A. Vincent[5], Syma Khalid[4], Gustav Berggren[3] & Chris Greening[1,12,13,14✉]

Diverse aerobic bacteria use atmospheric $H_2$ as an energy source for growth and survival[1]. This globally significant process regulates the composition of the atmosphere, enhances soil biodiversity and drives primary production in extreme environments[2,3]. Atmospheric $H_2$ oxidation is attributed to uncharacterized members of the [NiFe] hydrogenase superfamily[4,5]. However, it remains unresolved how these enzymes overcome the extraordinary catalytic challenge of oxidizing picomolar levels of $H_2$ amid ambient levels of the catalytic poison $O_2$ and how the derived electrons are transferred to the respiratory chain[1]. Here we determined the cryo-electron microscopy structure of the *Mycobacterium smegmatis* hydrogenase Huc and investigated its mechanism. Huc is a highly efficient oxygen-insensitive enzyme that couples oxidation of atmospheric $H_2$ to the hydrogenation of the respiratory electron carrier menaquinone. Huc uses narrow hydrophobic gas channels to selectively bind atmospheric $H_2$ at the expense of $O_2$, and 3 [3Fe–4S] clusters modulate the properties of the enzyme so that atmospheric $H_2$ oxidation is energetically feasible. The Huc catalytic subunits form an octameric 833 kDa complex around a membrane-associated stalk, which transports and reduces menaquinone 94 Å from the membrane. These findings provide a mechanistic basis for the biogeochemically and ecologically important process of atmospheric $H_2$ oxidation, uncover a mode of energy coupling dependent on long-range quinone transport, and pave the way for the development of catalysts that oxidize $H_2$ in ambient air.

The oxidation of atmospheric hydrogen ($H_2$) by soils is a key biogeochemical process that shapes the redox state of the atmosphere[1]. Until recently, this was thought to be an abiotic process, but it is now recognized that diverse aerobic bacteria from at least nine phyla oxidize atmospheric $H_2$ and together account for 75% (around 60 Tg) of the total $H_2$ removed from the atmosphere annually[1,4,6]. Atmospheric $H_2$ oxidation provides bacteria with a supplemental energy source in nutrient-limited soil environments, enabling them to either grow mixotrophically[7–11] or persist on air alone in a dormant but viable state for long periods[2,4,6,12–15]. For example, *Mycobacterium* cells and *Streptomyces* spores survive starvation by transferring electrons through an aerobic respiratory chain from atmospheric $H_2$ to $O_2$ (refs. [7,14,16,17]). The ability to oxidize atmospheric $H_2$ is widespread in bacteria from diverse environments[2], and some ecosystems—such as hyper-arid

polar soils—appear to be driven primarily by atmospheric energy sources[1–3,15].

There are no known chemical catalysts that can oxidize atmospheric $H_2$; this would require the selective oxidation of low concentrations of substrate (530 parts per billion by volume (ppbv)) present in an atmosphere containing a high concentration (21%) of the catalytic poison $O_2$ (refs. [18,19]). Group 1 [NiFe] hydrogenases constitute a family of membrane-bound $H_2$-oxidizing metalloenzymes that support aerobic and anaerobic growth of bacteria in $H_2$-rich environments; however, these enzymes are generally incapable of atmospheric $H_2$ oxidation, given they have a low affinity for $H_2$ (Michaelis constant ($K_m$) >500 nM) and are either reversibly or irreversibly inhibited by $O_2$ (refs. [20–24]). Recently, several high-affinity lineages of group 1 and 2 [NiFe] hydrogenases have been identified that input atmospheric $H_2$-derived electrons into the aerobic

[1]Department of Microbiology, Biomedicine Discovery Institute, Monash University, Clayton, Victoria, Australia. [2]Ramaciotti Centre for Cryo-Electron Microscopy, Monash University, Clayton, Victoria, Australia. [3]Department of Chemistry, Ångström Laboratory, Uppsala University, Uppsala, Sweden. [4]Department of Biochemistry, University of Oxford, Oxford, UK. [5]Department of Chemistry, University of Oxford, Inorganic Chemistry Laboratory, Oxford, UK. [6]Department of Physics, Experimental Molecular Biophysics, Freie Universität Berlin, Berlin, Germany. [7]Department of Biochemistry, Monash Biomedicine Discovery Institute, Monash University, Clayton, Victoria, Australia. [8]Monash Proteomics and Metabolomics Facility, Monash Biomedicine Discovery Institute, Monash University, Clayton, Victoria, Australia. [9]Centre for Electron Microscopy of Membrane Proteins, Monash Institute of Pharmaceutical Sciences, Parkville, Victoria, Australia. [10]Department of Chemistry and Biochemistry, The Ohio State University, Columbus, OH, USA. [11]Department of Microbiology and Immunology, University of Otago, Dunedin, New Zealand. [12]Securing Antarctica's Environmental Future, Monash University, Clayton, Victoria, Australia. [13]Centre to Impact AMR, Monash University, Clayton, Victoria, Australia. [14]ARC Research Hub for Carbon Utilisation and Recycling, Monash University, Clayton, Victoria, Australia. [15]These authors contributed equally: Rhys Grinter, Ashleigh Kropp. ✉e-mail: rhys.grinter@monash.edu; chris.greening@monash.edu

respiratory chain[3–6]. Whole-cell studies suggest that these enzymes have significantly higher apparent affinities for $H_2$ ($K_m$ values 30 to 200 nM) and appear insensitive to inhibition by $O_2$ (refs. [4,6,10,25,26]). However, given these hydrogenases have yet to be isolated, it remains unknown how they have evolved to selectively oxidize $H_2$, tolerate exposure to $O_2$ and interact with the electron transport chain. Notably, it is debated whether the hydrogenases responsible for the oxidation of atmospheric $H_2$ have an inherently high affinity or whether their affinity is modulated by their interactions with the respiratory chain[1,23].

To address these knowledge gaps, we investigated the structural and mechanistic basis of atmospheric $H_2$ oxidation in the aerobic bacterium *M. smegmatis*. This bacterium possesses 2 phylogenetically distinct hydrogenases—Huc (group 2a) and Hhy (group 1h)—that both oxidize $H_2$ to sub-atmospheric levels[4,17,26]. We isolated Huc directly from *M. smegmatis* and determined the structural and biochemical basis for its oxidation of atmospheric $H_2$.

## Huc reduces menaquinone analogues in air

We isolated Huc from *M. smegmatis* using a chromosomally encoded Strep-tag II on its HucS subunit. Analysis by SDS–PAGE showed that Huc consists of three protein subunits corresponding to HucL (approximately 58 kDa), HucS–2×Strep (approximately 39 kDa) and a third unknown subunit (approximately 18 kDa) (Extended Data Fig. 1a). Mass spectrometry identified this third subunit as the uncharacterized protein MSMEG_2261, which is encoded by an open reading frame in the *huc* operon directly upstream of *hucS*[17] (Extended Data Fig. 1b). The co-purification of MSMEG_2261 with the large and small hydrogenase subunits suggests that it is an additional component of the Huc complex, which we designate HucM. As previously observed for Huc in *M. smegmatis* cell lysates, the purified Huc complex migrates on a native gel at a molecular mass of around 800–900 kDa (Extended Data Fig. 1c)[26]. Huc is red–brown in colour, consistent with the presence of multiple metal clusters, and is highly stable at room temperature with a melting temperature of 78.3 °C (Extended Data Fig. 1d). To investigate the role of HucM in the Huc complex, we deleted the *hucM* gene from *M. smegmatis*. Activity staining of cell lysates shows that although Huc is active in this mutant, the high molecular weight species was not present, suggesting that HucM is critical for the formation of the Huc oligomer (Extended Data Fig. 1e).

Although Huc can oxidize atmospheric $H_2$ in whole cells, it was unclear whether it has an inherently high substrate affinity or if this property results from coupling to the respiratory electron transport chain[4]. To resolve this question, we tested the ability of purified Huc to oxidize 3 to 100 parts per million (ppm) $H_2$ in an ambient air headspace. Huc rapidly consumed $H_2$ to below the limit of detection of the gas chromatograph (about 40 ppbv) using nitro blue tetrazolium (NBT) or the quinone analogue menadione as the electron acceptor (Fig. 1a,b and Extended Data Fig. 1f). By contrast, the model $O_2$-tolerant group 1d [NiFe] hydrogenase Hyd1 from *Escherichia coli*[27,28] operated more slowly under these conditions and was unable to oxidize $H_2$ to below 4 parts per million by volume (ppmv) (Fig. 1b and Extended Data Fig. 1f). Kinetic analysis of purified Huc indicates that it is adapted to oxidize atmospheric $H_2$, with a high affinity ($K_m$ = 129 nM) and low $H_2$ threshold (<31 pM) but slow turnover (catalytic constant ($k_{cat}$) = 7.05 s⁻¹) (Fig. 1a,d and Supplementary Table 1). The $k_{cat}/K_m$ of Huc is $5.4 \times 10^7$ M⁻¹ s⁻¹, indicating the enzyme is highly efficient at low $H_2$ concentrations. The kinetics of purified Huc are similar to that of *M. smegmatis* mutants harbouring Huc as the sole hydrogenase ($K_m$ = 184 nM, threshold = 133 pM), suggesting that the hydrogenase itself is the primary determinant of whole-cell $H_2$ affinity[4]. The ability of Huc to oxidize $H_2$ at low concentrations in ambient air also indicates a high tolerance to inhibition by $O_2$. To assess the extent of Huc $O_2$ tolerance, we amperometrically measured its rate of $H_2$ oxidation in buffer containing $O_2$ at 0%, 10% (around 0.12 mM) and 100% (around 1.2 mM) saturation. We found

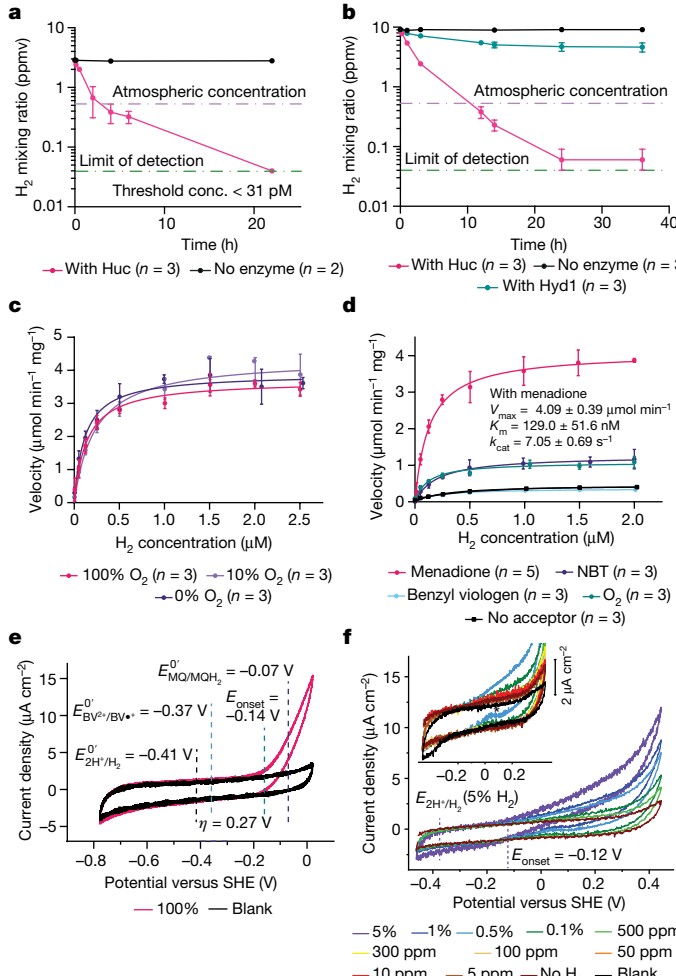

**Fig. 1 | Huc is a high-affinity $O_2$-insensitive hydrogenase. a**, Gas chromatography analysis of the $H_2$ concentration of the headspace of sealed vials containing Huc or no enzyme, with 200 μM menadione as the electron acceptor. Huc can oxidize $H_2$ below atmospheric concentrations (violet line) to the limit of detection (green line). conc., concentration. **b**, Gas chromatography analysis of vials containing Huc, *E. coli* Hyd1, or no enzyme, with 200 μM NBT as the electron acceptor. Huc—but not Hyd1—can oxidize $H_2$ at sub-atmospheric levels. **c**, Michaelis–Menten kinetics of $H_2$ oxidation by Huc in buffer with different percentages of $O_2$ saturation, with 200 μM menadione (the 0% $O_2$ comprises degassed buffer with traces of $O_2$). **d**, Michaelis–Menten kinetics of Huc $H_2$ consumption in buffer with different electron acceptors, degassed but with traces of $O_2$. In **a**–**d**, data are mean ± s.d. $V_{max}$, maximum velocity. **e**, Cyclic voltammogram of an immobilized Huc protein film in a 100% $H_2$ atmosphere. $E_{onset}$ is the Huc onset potential determined from the average of four experiments (a single representative curve is shown for clarity). $\eta$ is the estimated Huc overpotential compared to the $2H^+/H_2$ redox couple potential ($E^{0'}_{2H^+/H_2}$). The midpoint potentials of benzyl viologen ($E^{0'}_{BV^{2+}/BV^{•+}}$) and menaquinone ($E^{0'}_{MQ/MQH_2}$) are indicated. SHE, standard hydrogen electrode. **f**, Cyclic voltammogram of an immobilized Huc protein film with a gas mixture flushing through the headspace comprising argon, $N_2$ and the indicated level of $H_2$. The dashed vertical lines indicate the potential of the $2H^+/H_2$ couple and Huc onset potential, calculated for 5% $H_2$ at ambient pressure. Scans were recorded in the order of low to high $H_2$. The inset plot shows a zoomed view of the voltage response of Huc at low $H_2$ partial pressures.

no significant differences in the rate or affinity of Huc-mediated $H_2$ oxidation with varying $O_2$ concentrations, indicating that Huc is an $O_2$-insensitive enzyme (Fig. 1c). In the absence of an alternative electron acceptor, $O_2$ stimulates $H_2$ oxidation by Huc (Fig. 1d), indicating the enzyme can transfer electrons directly to $O_2$.

In *M. smegmatis* cells, electrons from $H_2$ oxidation by Huc enter the respiratory electron transport chain via the reduction of menaquinone, the major respiratory quinone of mycobacteria[26], but it is not known whether this reduction occurs directly or via an unknown membrane protein intermediate. To test whether menaquinone is the immediate electron acceptor of Huc, we determined its enzyme kinetics with either menadione (representing the redox head group of menaquinone) or the redox-active dyes NBT or benzyl viologen as electron acceptors. Consistent with menaquinone acting as its physiological electron acceptor, Huc displayed a much higher maximum velocity with menadione ($E^{0'} = -74$ mV) as its electron acceptor than NBT ($E^{0'} = -80$ mV), despite their similar midpoint potentials[29,30] (Fig. 1d and Supplementary Table 1). The low-potential electron acceptor benzyl viologen ($E^{0'} = -374$ mV) did not induce $H_2$ oxidation compared with the buffer control, indicating it does not accept electrons from Huc[31] (Fig. 1d). This is consistent with the role of Huc in oxidizing $H_2$ at low concentrations, as the higher redox potential of the $2H^+/H_2$ redox couple under ambient conditions ($E^{0'} = -136$ mV) would preclude the reduction of low-potential substrates. This suggests Huc is tuned to higher redox potentials.

To probe the potential dependence of Huc activity, we performed cyclic voltammetry on Huc films over a wide range of $H_2$ partial pressures (Fig. 1e,f). At lower partial pressures of $H_2$, the electrocatalytic current is observable above the background capacitive current down to 10 ppm $H_2$ (Fig. 1f). Huc maintained an overpotential of $255 \pm 10$ mV over the range of $H_2$ partial pressures tested (Extended Data Fig. 2a). For example, at 1 µM $H_2$, at which menadione reduction is close to substrate saturation by $H_2$, the onset potential ($E_{onset}$) measured for Huc is $-80$ mV $\pm$ 10 mV, just negative of the menadione potential ($E^{0'} = -74$ mV), which itself would be sensitive to the ratio of oxidized to reduced species. This is consistent with the role of Huc in oxidizing $H_2$ at low concentrations and would provide selectivity for the reduction of menaquinone over lower potential electron acceptors. Huc shows the greatest overpotential for a [NiFe] hydrogenase recorded to date[27,32–34] (see Extended Data Fig. 2b,c for comparison to *E. coli* Hyd1 and Hyd2). No significant current attributable to $H^+$ reduction by Huc was observed, and this catalytic irreversibility is consistent with the enzyme's high overpotential for $H_2$ oxidation (Fig. 1e). At higher levels of $H_2$, Huc does not show evidence of oxidative inactivation or reductive re-activation that are characteristic of other [NiFe] hydrogenases (Extended Data Fig. 2b,c), although there is evidence of a slight re-activation peak at 0.5% $H_2$ (indicated by the asterisk in Fig. 1e, inset).

## Huc oligomerizes around a central stalk

To determine the structure of Huc, we performed cryo-electron microscopy (cryo-EM) imaging and single-particle reconstruction of the purified 833 kDa complex. Cryo-EM micrographs and resulting class averages revealed a molecule shaped like a four-leaf clover, which is sometimes associated with membrane vesicles via a stalk-like protrusion (Extended Data Fig. 3a,b). In the Huc-overexpressing strain used for purification (see Methods for details), Huc activity is associated mainly with the soluble fraction, yet previous reports indicate that Huc predominantly associates with the cellular membrane[4,26], which is consistent with the presence of membrane-associated Huc in our sample. Three-dimensional reconstruction of the Huc complex yielded maps at a 2.19 Å overall resolution (Extended Data Fig. 3c–e and Extended Data Table 1), showing that each of the four lobes of Huc are composed of two Huc protomers, each consisting of a HucS and HucL subunit (Fig. 2a). The four Huc lobes are bound to a scaffold formed by four molecules of HucM, an elongated α-helical protein that intertwines to form a cage-like structure (Fig. 2b and Extended Data Fig. 4a,b). Because of the flexibility exhibited by this complex (Extended Data Fig. 3f and Supplementary Video 1), we performed 3D reconstruction of the individual Huc lobe from two datasets, obtaining resolutions of

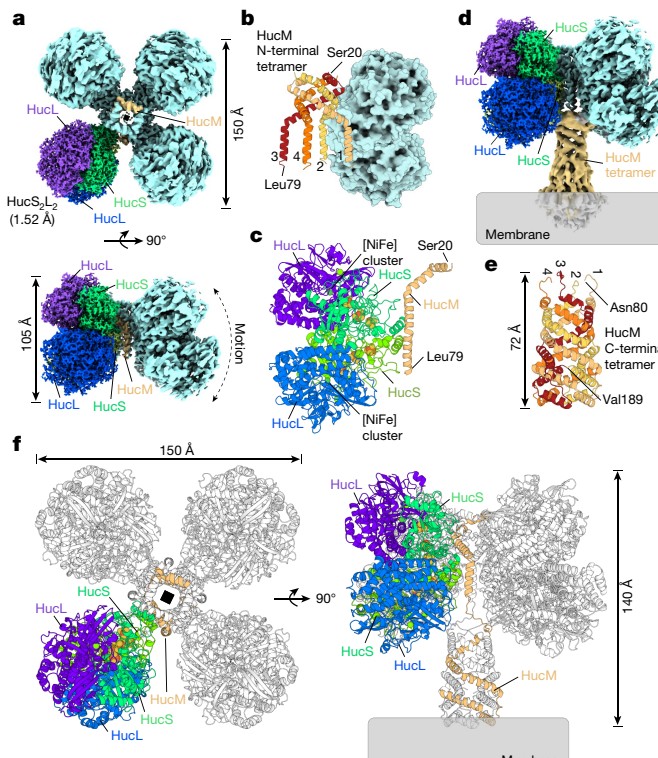

**Fig. 2 | Huc forms an 833-kDa oligomer comprising HucS, HucL and HucM. a**, Cryo-EM density maps of the Huc oligomer showing its four lobes that each contain two HucSL dimers, with four centrally located HucM subunits acting as a scaffold for the oligomer. One lobe of Huc shows the 1.52 Å high-resolution map, with HucL and HucS subunits coloured individually as indicated. One HucM molecule is coloured orange. **b**, A cartoon representation of the central tetramer formed by 4 HucM subunits (residues 20 to 79), with a single HucS$_2$L$_2$ lobe shown as a surface model for context. **c**, A cartoon representation of the asymmetric unit of the Huc oligomer. **d**, A composite cryo-EM density map showing the Huc 'body' region as in **a**, and the 'stalk' region that is peripherally associated with the cell membrane. **e**, A cartoon representation of the AlphaFold2 model of the C-terminal region of HucM (residues 80 to 189), showing that it forms a tetrameric coiled-coil tube. **f**, A cartoon representation showing the full Huc complex reconstructed from the cryo-EM structure and AlphaFold model of the HucM C-terminal fitted to the cryo-EM density map of the stalk region.

1.52 Å and 1.67 Å (Fig. 2c, Extended Data Fig. 4c,d, Extended Data Table 1 and Supplementary Figs. 2 and 3). The HucSL heterodimer is similar to that of other [NiFe] hydrogenases, with a root mean squared deviation (rmsd) of 3.7 Å and 2.8 Å to hydrogenases from *Desulfovibrio vulgaris* Miyazaki F (Protein Data Bank (PDB) ID = 4U9H) and *Cupriavidus necator* H16 (PDB ID 5AA5), respectively[23,24]. Consistently, the architecture of the [NiFe] active site is the same as previously reported[35] (Fig. 3a). Density maps and the fourth coordinating residue of the 3 iron–sulfur clusters of HucS indicate they all have a [3Fe–4S] configuration (Fig. 3b, Extended Data Fig. 5a and Supplementary Videos 2–4). This is supported by electron paramagnetic resonance (EPR) analysis of Huc, which yields a signal consistent with the presence of multiple spin-coupled [3Fe–4S] clusters and demonstrates their redox activity (Fig. 3c, Extended Data Fig. 5b and Supplementary Note 1). This is in contrast to other characterized hydrogenases, including $O_2$-tolerant variants, that possess one or more [4Fe–4S] clusters[21,35]. [3Fe–4S] clusters generally undergo their +/0 redox transition at approximately +250 mV higher potential than [4Fe–4S]$^{2+/+}$ clusters[36]. As the potential of the electron transfer chain has previously been shown to dictate catalytic bias in other hydrogenases, the presence of exclusively [3Fe–4S] clusters in Huc probably has a major role in tuning the overpotential of the enzyme[37].

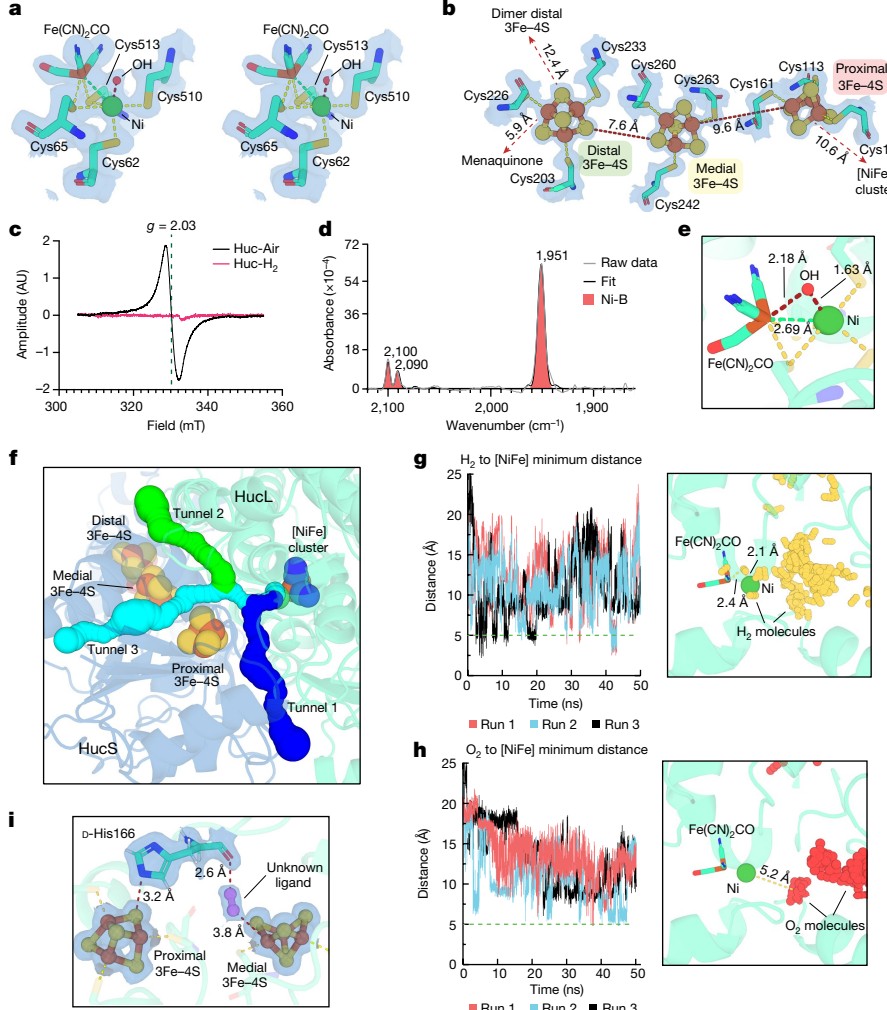

**Fig. 3 | Huc adopts a non-inhibitory Ni-B state under ambient conditions and restricts O₂ diffusion to the catalytic cluster. a**, A stereo view of the Huc [NiFe] cluster under ambient air. **b**, The structure of the three HucS [3Fe–4S] clusters. **c**, High-temperature (30K) EPR spectra of Huc in the as-isolated oxidized state (Huc-Air) and the $H_2$-reduced state (Huc-$H_2$). The isotropic signal at $g$ = 2.03 ($g$ is the Landé factor) is associated with the oxidized [3Fe–4S]⁺ clusters, which disappears in Huc-$H_2$ owing to reduction of the clusters. No signal attributable to reduced [4Fe–4S]⁺ clusters is discernible in the $H_2$-treated form of Huc. **d**, FTIR spectra of oxidized Huc isolated in ambient air, showing that the [NiFe] active site of the enzyme adopts an oxidized state consistent with an Ni-B state. **e**, The Huc [NiFe] active site, showing the relative distances between the Ni-B state hydroxyl ligand and the Ni and Fe ions of the cluster. **f**, The HucSL dimer, showing the location and width of the hydrophobic gas channels that provide substrate access to the [NiFe] active site. **g**, Left, plot of distance versus time, showing the closest distance between $H_2$ and the Huc [NiFe] active site during molecular dynamics simulations of HucSL in the presence of $H_2$. Right, the location of $H_2$ molecules in a representative subset of simulation frames when $H_2$ molecules are in the closest proximity to the active site. **h**, Similar to **g**, but shows location of O₂ relative to the Huc [NiFe] site in simulations of HucSL performed in the presence of O₂. **i**, The proximity of HucL D-His166 to the proximal and medial [3Fe–4S] clusters and the location of the unknown ligand. Cryo-EM density potential maps in **a**,**b**,**i** are shown as a transparent blue surface at 5σ.

Out of the 188 amino acids of HucM, only residues 20–79 could be modelled into the cryo-EM density maps, with diffuse density corresponding to the remaining C-terminal region of HucM forming the stalk-like protrusion. To better resolve this region of the structure, we masked it and performed signal subtraction on the remainder of the Huc structure, followed by focused refinement and 3D classification (Extended Data Table 1 and Supplementary Fig. 4). This processing showed that the C-terminal region of HucM forms a tube connecting the main body of the Huc oligomer to the membrane (Fig. 2d and Extended Data Fig. 4e,g). AlphaFold modelling shows this tube consists of a symmetrical coiled-coil of α-helices composed of the four HucM molecules that encloses a chamber at the centre of Huc (Fig. 2e and Extended Data Fig. 4f,g). These α-helices are amphipathic, with the hydrophobic face pointing inwards, lining the tube entirely with hydrophobic residues (Extended Data Fig. 4h). Interaction of the tube with the phospholipid head groups of the membrane is facilitated by an outward-facing helix at the tube terminus containing arginine, lysine and tryptophan residues (Extended Data Fig. 4i). By docking the model of the HucM tetramer into the map for the Huc stalk, we resolved the structure of the complete Huc complex (Fig. 2f and Supplementary Video 5).

## Unique features enable ambient H₂ catalysis

Huc is inherently capable of $H_2$ oxidation at sub-atmospheric concentrations and is largely insensitive to inhibition by O₂ (Fig. 1c). This is remarkable for a hydrogenase and highly desirable for the development of biocatalysts that oxidize $H_2$ under aerobic conditions[38,39]. Although Huc may use reverse electron flow to reduce O₂ bound at the active site to $H_2O$, it cannot behave as previously described O₂-tolerant hydrogenases (for example, *E. coli* Hyd-1), as it lacks a proximal [4Fe–3S] cluster

to enable the rapid transfer of four electrons[21,22,40] (Supplementary Note 2). As in several other hydrogenases[23,41], the distal Fe–S clusters of the $HucS_2L_2$ dimer are within electron transfer distance (Figs. 2c and 3b), which could provide a source of electrons from $H_2$ oxidation by the neighbouring dimer subunit to reduce bound $O_2$ (refs. [23,41]). However, as Huc operates under atmospheric conditions, where $O_2$ is four million times more abundant than $H_2$ (refs. [18,19]), it is unlikely that $H_2$ oxidation alone would keep the Huc dimers sufficiently reduced to sustainably displace bound $O_2$, meaning the enzyme must resist $O_2$ inhibition by additional mechanisms. Concordantly, analysis of the hydrophobic gas channels of Huc that provide $H_2$ access to the active site indicates that they are markedly narrower than those of structurally characterized $O_2$-sensitive and $O_2$-tolerant [NiFe] hydrogenases (Fig. 3f, Extended Data Fig. 6a and Supplementary Table 2). Instead, they are of similar width to the $O_2$-insensitive but low-affinity group 1h [NiFe] hydrogenase from *C. necator*[23] (Supplementary Table 2). To determine whether these channels have a role in sterically excluding $O_2$ from the Huc active site, we performed all-atom molecular dynamics simulations on the HucSL dimer in the presence of excess $H_2$ and $O_2$ dissolved in water. Of note, during these simulations $H_2$ enters the Huc active site, whereas $O_2$ is sterically excluded by a series of bottlenecks between the active site and enzyme surface and does not get closer than 5 Å to the catalytic cluster (Fig. 3g,h and Extended Data Fig. 6). The simulations suggest that the critical point for selection against $O_2$ is a bottleneck after the convergence of the three gas tunnels immediately preceding the active site entrance. In support of this observation, $O_2$ reached the active site in simulations of Huc variants with mutations that relieve this bottleneck (Extended Data Fig. 7 and Supplementary Note 3). This suggests that, as previously proposed for [NiFe] hydrogenases[42], the Huc gas channels have a role in protecting the [NiFe]-cluster from inactivation by $O_2$.

Although the narrow hydrophobic gas channels greatly kinetically favour $H_2$ diffusion over $O_2$, Fourier transform infrared (FTIR) spectroscopy analysis and the cryo-EM structure indicate that the active site of Huc isolated in ambient air is in an oxidized state with a bridging hydroxide ligand (Fig. 3a,d and Supplementary Note 2). The FTIR vibrational frequencies are consistent with an electron-deficient Ni-B species (Extended Data Fig. 8a,b). This is further supported by EPR spectra collected of air-exposed Huc, which shows a rhombic signal that closely resembles the canonical Ni-B species[32] (Extended Data Fig. 5c,d and Supplementary Note 1). Oxidation of the enzyme by $O_2$ resulting in this Ni-B state probably occurs via an indirect pathway, in line with previous studies showing the $O_2$-independent formation of Ni-B states[43,44]. The FTIR spectra support this hypothesis: when reduced Huc is exposed to an atmosphere containing 20% $O_2$, the Ni-SI state is rapidly populated at the expense of more reduced Ni-R and Ni-C states, followed by the slow appearance of the hydroxide-bound Ni-B (Extended Data Fig. 8e,f and Supplementary Note 2).

The high-affinity kinetics of Huc differ radically from more classical $O_2$-sensitive hydrogenases, which tend to be faster low-affinity enzymes[4,45] (Fig. 1d). A comparison of the environment surrounding the [NiFe] active site of Huc and $O_2$-sensitive hydrogenases reveals no differences that could readily explain the divergent properties of these enzymes (Extended Data Fig. 9a–c). This indicates that the overall activity of the [NiFe] active site instead results from changes in the properties of the iron–sulfur clusters of the small subunit or other regions of the enzyme. Given their distinct structure, the modified gas channels identified in Huc may have a role in selectively capturing $H_2$ molecules and delivering them to the active site, decreasing the rate but increasing the enzyme's affinity[42]. In addition, the increased redox potential of Huc ($E_{onset} \approx -80$ to $-160$ mV) (Fig. 1e,f) means that atmospheric $H_2$ oxidation, with a redox potential ($E_{atm}$) of $-0.134$ mV at pH 7, is thermodynamically favourable in comparison to low-affinity variants operating at minimal overpotential[46] ($E_{onset} \approx -360$ mV). The redox properties of the Huc proximal and medial clusters may be further modified by an unusual D-isomer of histidine, indicated by the

cryo-EM maps, at position 166 of HucL (Extended Data Fig. 9d and Supplementary Videos 6 and 7). An L-isomeric histidine is conserved at this position in other hydrogenases, with its imidazole head group interacting with the proximal [Fe–S] cluster (Extended Data Fig. 9e). In Huc, His166 also interacts with the proximal cluster (Fig. 3i), but the D-isomeric configuration shifts the position of the backbone carbonyl of His166 so that it interacts with a highly coordinated ligand in proximity to the medial cluster (Fig. 3i and Extended Data Fig. 9f). We were unable to identify this ligand, although its elongated density suggests that it is not water, and it may have a role in tuning the redox potential of the medial [3Fe–4S] cluster.

## Huc reduces menaquinone internally

Analysis of the Huc EM maps reveals additional density near the distal [3Fe–4S] cluster of each Huc protomer, which could be unambiguously modelled as menaquinone (Fig. 4a,b and Supplementary Video 8). Consistent with this, mass spectrometry confirmed that β-dihydromenaquinone-9 associates with purified Huc with an occupancy of 96.2% (Fig. 4e and Supplementary Fig. 5). The head group of menaquinone is stabilized by hydrogen bonding between its oxygen groups and the terminal hydroxyl of Tyr301 and the backbone carbonyl of Lys212, as well as π–π interactions with Tyr229 of HucS (Fig. 4c). In our 1.67 Å reconstruction, Tyr229 adopts a second conformation that clashes with density corresponding to menaquinone, which shields the proximal [3Fe–4S] cluster from the solvent and may mitigate $O_2$ reduction in the absence of menaquinone, which would lead to the formation of reactive oxygen species (Fig. 4d and Supplementary Video 9). These data indicate that Huc directly binds and reduces menaquinone, which is consistent with menadione acting as the preferred electron acceptor for Huc (Fig. 1c). This contrasts with observations made in group 1 [NiFe] hydrogenases, where small subunits are electron relays without catalytic roles[20–24].

However, despite this strong evidence that Huc directly donates electrons from $H_2$ oxidation to respiratory quinone, Huc is not an integral membrane protein, and thus the highly hydrophobic, membrane-bound menaquinone must be transported to its substrate-binding site 94 Å away from the membrane surface. Analysis of the structure of the Huc oligomer indicates a possible mechanism by which this could occur. The interior surface of the internal cavity of Huc is lined by hydrophobic sidechains of HucS and HucM, which facilitate the binding of menaquinone and other hydrophobic molecules and has dimensions capable of accommodating menaquinone (Fig. 4f–h, Extended Data Fig. 4h and Supplementary Video 10). The conduit for menaquinone entry to and exit from the hydrophobic chamber is probably provided by the membrane-associated Huc stalk. Overall, menaquinone molecules probably diffuse through this tube into the Huc hydrophobic chamber, where they bind the substrate acceptor site and are reduced to menaquinol before diffusing back into the membrane, where they are oxidized by the cytochrome $bcc$-$aa_3$ oxidase terminal oxidase to generate proton motive force[26] (Fig. 4i). Among hydrogenases, this mechanism of quinone reduction is unique to Huc, with the other structurally characterized respiratory hydrogenases reducing quinone in the membrane via direct electron transfer to an integral membrane cytochrome $b$ subunit[28]. There are, however, some similarities with the reduction mechanism of respiratory chain complex I, which removes quinone approximately 30 Å from the membrane to position it for reduction by a [4Fe–4S] cluster within its soluble arm; however, in complex I[47,48], this extraction is achieved via an enclosed channel rather than the broader hydrophobic tube and chamber observed in Huc[47].

## Conclusions

To use the trace quantities of $H_2$ present in air, the [NiFe] hydrogenases of trace-gas-scavenging bacteria require distinct properties compared

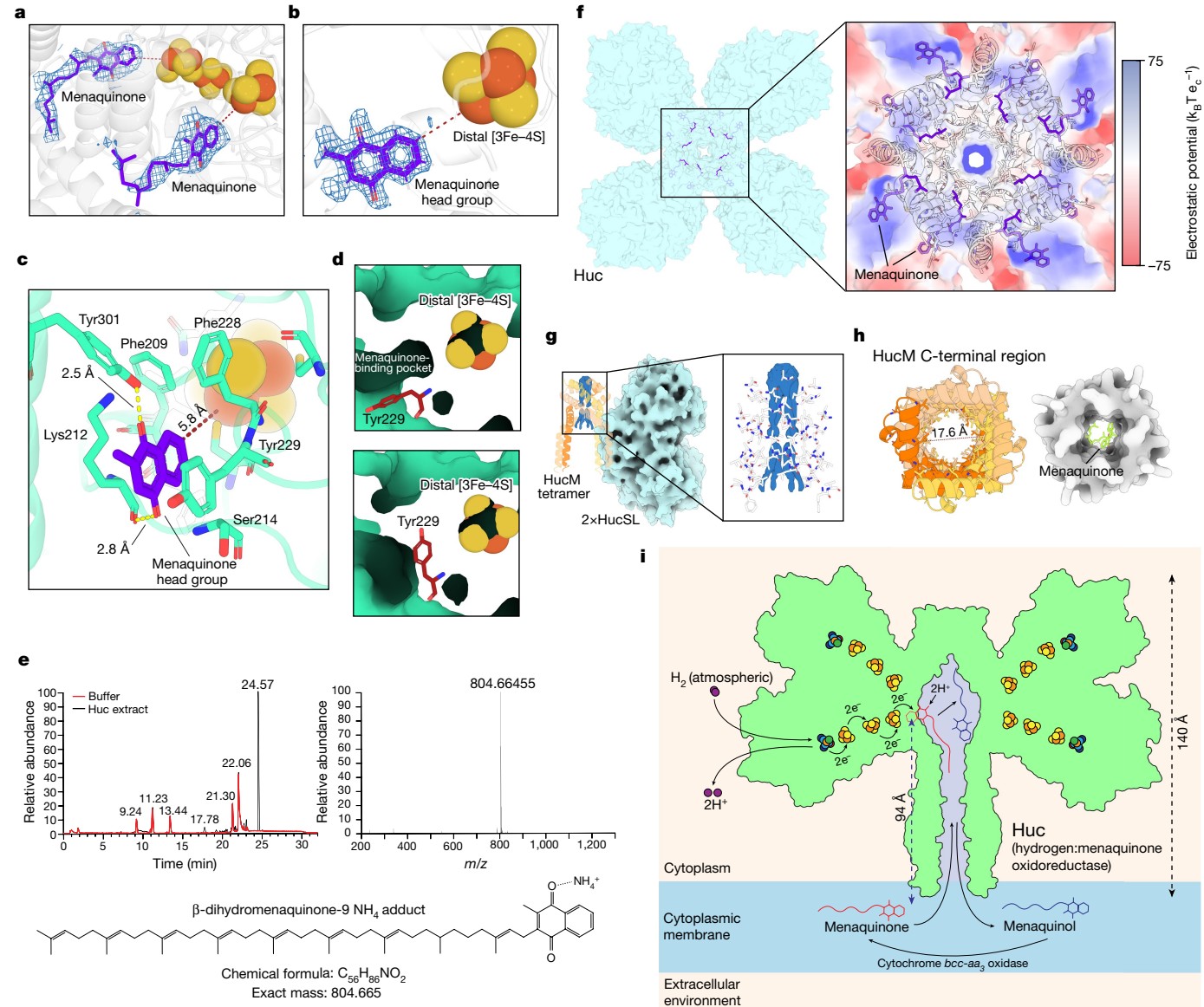

**Fig. 4 | Huc extracts menaquinone from the membrane and directly reduces it. a**, Density corresponding to menaquinone bound to Huc from low-resolution cryo-EM reconstruction of the Huc oligomer. **b**, Density corresponding to bound menaquinone in high-resolution cryo-EM maps of the Huc dimer. Density for the menaquinone tail is absent in these maps owing to masking and symmetry averaging. **c**, Coordination of the menaquinone head group by the HucS subunit within electron transfer distance of the distal [3Fe–4S] cluster. **d**, The two conformations observed for Tyr229 of HucS in the high-resolution Huc cryo-EM density maps. Tyr229 adopts an open conformation in the presence of menaquinone (top) and a closed conformation (bottom) that is mutually exclusive with menaquinone binding. **e**, High performance liquid chromatography–mass spectrometry (LC–MS) analysis of the Folch extract

from purified Huc. The base peak chromatogram (left) shows a substantial peak at 24.6 min corresponding to an ion at $m/z = 804.66455$ (right), which is consistent with the ammonium adduct of β-dihydromenaquinone-9, $m/z = 804.66531$ (bottom). **f**, A hybrid stick and electrostatic surface representation of the central cavity Huc, showing that the surface of the chamber is lined with hydrophobic residues. **g**, A view of the HucM tetramer, showing the presence of density for lipids that occlude the top of the Huc internal hydrophobic chamber. **h**, A top-down view of the HucM C-terminal tube, showing the interior of the tube is lined with hydrophobic residues (left) and can accommodate a menaquinone molecule (right). **i**, A model for the reduction of menaquinone by Huc, using electrons derived from the oxidation of atmospheric $H_2$.

with their counterparts that function under $H_2$-rich hypoxic or anoxic conditions[4,35]. Using biochemical and electrochemical characterization of Huc from the aerobic bacterium *M. smegmatis*, we show that the required properties of $O_2$ insensitivity and high affinity for $H_2$ are inherent to this hydrogenase, rather than resulting from coupling to other processes inside the bacterial cell. Moreover, through determination of the cryo-EM structure of Huc, molecular dynamics simulations, and FTIR and EPR spectroscopy, we provide strong evidence that at least partial exclusion of $O_2$ from the active site contributes

to the $O_2$ insensitivity of the enzyme. Our data show that Huc has a large electrochemical overpotential that makes it uniquely tuned for the oxidation of trace quantities of $H_2$ and for the direct donation of the resulting electrons to the respiratory cofactor menaquinone. Of note, we demonstrate that Huc accesses menaquinone via a unique and highly unusual mechanism. Through the scaffolding protein HucM, the Huc complex can extract menaquinone from the membrane and transport it 94 Å to the electron acceptor site of the enzyme. This finding greatly expands our knowledge of the possibilities for performing respiratory

quinone reduction. More broadly, these findings open pathways for biocatalyst development given that all hydrogenases applied so far in whole-cell and purified enzyme systems are low-affinity enzymes that become inhibited by $O_2$. Huc, as an oxygen-insensitive high-affinity enzyme and a group 2 [NiFe] hydrogenase, provides a basis for the development of biocatalysts that operate under ambient conditions.

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

## Methods

### Statistics and reproducibility

**Huc purification and PAGE analysis.** Huc was purified from *M. smegmatis* cells on eight separate occasions throughout the course of the study. The yield of purified Huc (mg per litre of culture) varied by a factor of five between purifications. The relative abundance of the Huc oligomer and low molecular weight species (Extended Data Fig. 1a) varied approximately twofold between purifications. SDS–PAGE and activity staining was performed on Huc from each purification ($n = 8$). The abundance of HucS, HucL and HucM subunits in the Huc oligomer (Extended Data Fig. 1b) and its activity (Extended Data Fig. 1c) was consistent between purifications, so gels from a single purification were shown. Huc oligomer from different purifications was utilized for downstream enzymatic, electrochemical and spectroscopic analysis. Statistical methods were not used to predetermine sample size for this study. Blinding and randomization were not used as they are not appropriate for this kind of study.

**HucM mutant analysis.** Native PAGE analysis of Huc activity in wild-type and *ΔhucM* cell lysates was performed three times using cell lysates derived from independently grown cultures ($n = 3$ biological replicates). All replicates displayed comparable relative activity and band sizes. A single representative gel is shown (Extended Data Fig. 1e).

**Huc differential scanning fluorimetry.** Differential scanning fluorimetry experiments to determine the Huc melting temperature were performed three times on a single batch of purified Huc ($n = 3$ technical replicates) with consistent results between samples. Given the consistent purity and activity of the Huc oligomer, analysis of a single sample was sufficient to determine the melting temperature of the enzyme. Representative curves from a single experiment are shown (Extended Data Fig. 1d).

**Gas chromatography.** Huc $H_2$ consumption experiments were performed on 4 different Huc preparations with each electron acceptor (NBT or menadione) ($n = 4$, biological replicates) with 3 replicate vials used in each case ($n = 3$ technical replicates). The data obtained were consistent between both biological and technical replicates. *E. coli* Hyd1 $H_2$ consumption experiments were performed on a single enzyme preparation ($n = 1$ biological replicates), with 3 replicate vials used ($n = 3$ technical replicates). Figure 1a,b and Extended Data Fig. 1f show data from a single enzyme preparation with data collected concurrently for all samples presented. Data are presented as a mean of the 3 technical replicates ($n = 3$) with error bars showing s.d. Control vials containing no enzyme showed no $H_2$ consumption or loss and were included in each experiment with 2 or 3 technical replicates ($n = 2$ or 3).

**Hydrogen electrode experiments.** Huc oxygen tolerance and electron acceptor dependence experiments were performed 3 times on separate Huc purifications ($n = 3$, biological replicates) with 3 separate electrode traces collected for each $O_2$ concentration or electron acceptor for each Huc sample ($n = 3$, technical replicates). For menadione activity experiments, 5 technical replicates were performed due to increased instrument noise when using this compound ($n = 5$, technical replicates). The data obtained were consistent between both biological and technical replicates. The mean of the three replicates for each condition (five for menadione) from a single Huc sample is shown with error bars showing s.d. (Fig. 1c,d).

**Protein film electrochemistry.** Cyclic voltammograms of Huc in 100% $H_2$ (Fig. 1e) were recorded using a single enzyme preparation ($n = 1$ biological replicates). Two immobilized Huc films were recorded ($n = 2$ technical replicates), with each film scanned at least two times. The data obtained were consistent between technical replicates. Two control films with no immobilized enzyme were included in each experiment ($n = 2$ technical replicates), each scanned two times. Cyclic voltammetry for Huc, Hyd1 and Hyd2 in 10 ppm to 5% $H_2$ were performed on a single preparation of each enzyme ($n = 1$, biological replicates) with 3 different protein films of each ($n = 3$ technical replicates) (Fig. 1f and Extended Data Fig. 2). The Huc used in these experiments was from a different purification to that in the 100% $H_2$ experiments. It is common in protein film electrochemistry to find variation in absolute enzyme coverage between different films arising from minor differences in film preparation; therefore it was not reasonable to compare the absolute current from electrochemistry between different protein films. However, the trends in current in response to changes in $H_2$ partial pressure were consistent between technical replicates. For determination of onset potentials for Huc, the mean of three objective readings of onset potential for two Huc protein films was determined, as shown with error bars showing s.d. (Extended Data Fig. 2a).

**Cryo-EM imaging.** Cryo-EM imaging was performed on Huc from 2 separate purifications ($n = 2$ biological replicates). In total, 2,226 and 3,113 images were collected from each of the purifications with single representative images of free and membrane-associated Huc down in Extended Data Fig. 3a. Two-dimensional and three-dimensional reconstructions of were performed independently on each dataset.

**EPR experiments.** EPR experiments were performed on a single enzyme preparation ($n = 1$ biological replicates), incubated under either $H_2$ or Ar before flash freezing. Each individual power and temperature setting was recorded once ($n = 1$ technical replicate), and each spectrum shown represents the average of 4 scans (Fig. 3c, Extended Data Fig. 5b–d). Given the consistency of the power and temperature dependence when cross-correlated, this was sufficient for the conclusions drawn in the manuscript.

**FTIR experiments.** FTIR experiments were performed on a single enzyme preparation ($n = 1$ biological replicates). The population of reduced states upon $H_2$ exposure and the formation of Ni-B upon exposure to air was performed with at least three individual attenuated total reflectance (ATR)-FTIR spectroscopy sample preparations ($n = 3$, technical replicates) leading to similar results (Fig. 3d and Extended Data Fig. 8a–d,h). The kinetics transferring between these redox state populations were recorded on one sample preparation ($n = 1$ technical replicates) (Extended Data Fig. 8e).

**Mass spectrometry.** The identification of dihydromenaquinone-9 in the purified Huc protein by LC–MS (Fig. 4e) was performed once ($n = 1$ technical replicate) due to the qualitative nature of the analysis and sample quantity constraints. While the estimate of occupancy (Supplementary Fig. 5) using these data is quantitative, the purpose of that analysis was predominantly to demonstrate that the signal was consistent with that anticipated for the ligand, which only required a single replicate.

**Molecular dynamics experiments.** Molecular dynamics simulations were performed 3 times for each condition simulated ($n = 3$). Cumulative gas occupancy plots are presented as the mean of these simulations with dotted lines showing s.d. (Extended Data Figs. 6b and 7b).

**Publicly available data.** Publicly available structural coordinates utilized in this study can be accessed via the following PDB accession codes, 5AA5, 2FRV, 4U9H, 6EHQ, 4C3O, 3AYX, 5MDK.

### Bacterial strains and growth conditions

*M. smegmatis* mc²155 and its derivatives were grown on lysogeny broth (LB) agar plates supplemented with 0.05% (w/v) Tween 80 (LBT). In broth culture, *M. smegmatis* was grown in either LBT, or Hartmans de Bont minimal medium supplemented with 0.2% (w/v) glycerol,

0.05% (w/v) tyloxapol, and 10 mm $NiSO_4$ (HdB). *E. coli* was maintained on LB agar plates and grown in LB broth. Selective LB or LBT medium used for cloning experiments contained gentamycin at 5 μg ml⁻¹ for *M. smegmatis* and 20 μg ml⁻¹ for *E. coli*. Cultures were incubated at 37 °C with rotary shaking at 150 rpm for liquid cultures unless otherwise specified. The strains of *M. smegmatis* and its derivatives and *E. coli* are listed in Supplementary Table 4.

## Genetic manipulation of *M. smegmatis*

For Huc production, protein expression was performed using the *M. smegmatis* mc²155 strain PRC1, which lacks the glycerol response regulator *gylR* (*gylR* Leu154→frameshift) and the Hhy hydrogenase (*hhyS* inactivation)[49,50]. To facilitate Huc purification, a 2×StrepII tag was inserted at the N-terminal end of the small subunit (MSMEG_2262, HucS) through allelic exchange mutagenesis as described previously[51]. An allelic exchange construct, hucS-2×StrepII (2656 bp), was synthesized by Genewiz and cloned into the SpeI site of the mycobacterial shuttle plasmid pX33 to yield the construct pHuc-2×StrepII (Supplementary Table 4). pHuc-2×StrepII was propagated in *E. coli* DH5α and transformed into wild-type *M. smegmatis* mc²155 PRC1 cells by electroporation. Gentamycin was used in selective solid and liquid medium to propagate pX33. To allow permissive temperature-sensitive vector replication, transformants were incubated on LBT gentamycin plates at 28 °C until colonies were visible (5–7 days). The resultant catechol-positive colonies were subcultured in LBT gentamycin broth at 40 °C for 3–5 days and then diluted from broth onto fresh LBT gentamycin plates and incubated at 37 °C for 3–5 days to facilitate the integration of the recombinant plasmid, via flanking regions, into the chromosome. The second recombination event was facilitated by subculturing catechol-reactive and gentamycin-resistant colonies in LBT supplemented with 10% sucrose (w/v) for 3–5 days, followed by dilution onto LBT agar plates supplemented with 10% sucrose (w/v) and incubating at 37 °C for 3–5 days. Gentamycin-sensitive and catechol-unreactive colonies were subsequently screened by PCR to distinguish wild-type revertants from HucS-2×StrII mutants. A colony that was PCR positive for the HucS-2×StrII insertion was designated *M. smegmatis* mc²155 PRC1 HucS-2×StrII and used for subsequent experiments. The primers used for screening are listed in Supplementary Table 4. HucM was deleted using allelic exchange mutagenesis using the same PRC1 parent strain and procedure for the 2×StrepII tagging of HucS. An allelic exchange construct consisting of ~1,000 bp of the genomic sequence on either side of HucM was used.

## Huc purification

*M. smegmatis* mc²155 PRC1 HucS-2×StrII cells were propagated in large volumes (10–20 l) of HdB until 24 h into stationary phase. Cells were harvested by centrifugation and resuspended in lysis buffer (50 mM Tris, 150 mM NaCl pH 8.0) supplemented with 0.1 mg ml⁻¹ lysozyme, 0.1 mg ml⁻¹ DNase, 1 mM $MgCl_2$ and protease inhibitor cocktail tablets. Cells were lysed by cell disruptor (Emulsiflex C-5), and cell debris were removed by centrifugation at 30,000$g$ for 15 min. Biotin blocking buffer (IBA) was added to clarified cell lysates (1 ml per 25 ml of lysate) before loading onto a 1 ml StrepTrap column (Cytiva), and the column was washed extensively with lysis buffer before bound Huc was eluted with lysis buffer + 2.5 mM desthiobiotin. Huc-containing fractions from StrepTrap were determined by SDS–PAGE, pooled, and concentrated using a 100 kDa MWCO centrifugal concentrator (Amicon, Millipore). Concentrated Huc was loaded on a Superose 6 10/300 column (Cytiva), fractions containing oligomeric Huc were confirmed by SDS- and native PAGE, pooled, and concentrated to ~6 mg ml⁻¹ before flash freezing in liquid $N_2$ and storage at −80 °C. Typically purification yielded 10–20 μg l⁻¹ of the Huc complex.

## PAGE analysis, activity staining and western blotting

For SDS–PAGE and native PAGE, samples were run on Bolt 4–12% SDS–PAGE and native PAGE 4-16% gels (Invitrogen) respectively, according to the manufacturer's instructions. Gels were stained for total protein by AcquaStain Protein Gel Stain (Bulldog) and hydrogenase activity using the colorimetric electron acceptor NBT hydrogenase. For NBT activity staining, gels were incubated in 50 mM Tris, 150 mM NaCl pH 8.0 supplemented with 500 μM NBT in an anaerobic jar amended with 7% $H_2$ anaerobic mix. Incubation was conducted for 30 mins to 12 h, depending on the level of activity. Bands exhibiting hydrogenase activity were identified based on the purple colour of reduced NBT. For western blotting, proteins were transferred to nitrocellulose membrane from SDS–PAGE or native PAGE gels using the Transblot Turbo semidry transfer apparatus (BioRad) at 25 V for 30 min. Following transfer, the protein-containing nitrocellulose membrane was blocked with 3% (w/v) BSA in PBS (pH 7.4) with 0.1% (v/v) Tween 20 (PBST). The nitrocellulose membrane was washed three times in 20 ml of PBST and finally resuspended in 10 ml of the same buffer. Strep-Tactin HRP conjugate was then added at a 1:100,000 dilution. Bound Strep-Tactin HRP was detected by chemiluminescence using the ECL Prime detection kit (Cytiva) following the manufacturer's specifications, blots were visualized using a ChemiDoc (BioRad).

## Amperometric measurement of Huc activity

$H_2$ consumption by purified Huc was measured amperometrically with a Unisense $H_2$ microsensor polarized to +100 mV for 1 h. Initially, the electrode was calibrated with known $H_2$ standards of 0%, 1%, and 10% $H_2$ (v/v) in gas saturated buffer (50 mM Tris, 150 mM NaCl, pH 8.0). This buffer was prepared by firstly bubbling all buffers with 100% nitrogen gas for 1 h to remove any traces of oxygen, then bubbling 100% $H_2$ (v/v) through the de-oxygenated buffer for 10 min. Before degassing and regassing, all buffers contained electron acceptors to a final concentration of 200 μM (menadione, NBT, or benzyl viologen). For each reading, 1 ml of 1% (8 μM) $H_2$-infused buffer containing the electron acceptor was added to the microrespiration assay chamber, to a final concentration of 1% $H_2$. The electrode was subsequently placed into the chamber to equilibrate. Once equilibrated (approximately 10 min), BSA (0.3 mg ml⁻¹) and Huc (1–3 nM in 0.3 mg ml⁻¹ BSA) were added to the chamber via a needle so as not to disrupt the gas balance in the solution. The changes in $H_2$ concentration were measured using the SensorTrace Suite v3.4.00 and linear rates of $H_2$ consumption were measured from the addition of Huc until complete $H_2$ consumption. The linear rate of $H_2$ consumption by Huc was calculated between a concentration of approximately 2.5 μM $H_2$ to 0.0125 μM $H_2$ over eight time points. These rates of $H_2$ consumption were used to plot a Michaelis-Menton curve and calculate Huc-specific activity in the presence of different electron acceptors, as well as the maximal velocity of Huc.

## Mass spectrometric menaquinone detection

Samples were prepared for LC–MS analysis using a modified Folch extraction. In brief, a 100 μl solution of purified protein (~57 μg) was treated with 2000 μl of 2:1 chloroform:methanol (v/v) after which the mixture was shaken for 10 min and allowed to stand for a further 50 min. Four-hundred microlitres of water was added and the mixture was shaken for 10 min after which the sample was allowed to stand until the two phases had completely separated. The lower chloroform-rich phase was then transferred to a 2 ml sample vial and the solvent was removed under a stream of nitrogen gas. The resulting residue was reconstituted in 100 μl of 2:1 chloroform:methanol v/v and transferred to a 200 μl sample insert, the solvent was again removed and the sample reconstituted in a 7:3 mixture of LC solvent A:LC solvent B (v/v). Samples were analysed using a Dionex RSLC3000 UHPLC (Thermo) coupled to a Q-Exactive Plus Orbitrap MS (Thermo) using a C18 column (Zorbax Eclipse Plus C18 Rapid Resolution HD 2.1 × 100 mm × 1.8 μm, Agilent) with a binary solvent system; solvent A = 40% isopropanol and solvent B = 98% isopropanol both containing 2 mM formic acid and 8 mM ammonium formate. Linear gradient time, %B as follows: 0 min, 0%; 8 min, 35%; 16 min, 50%; 19 min, 80%; 23 min, 100%; 28 min, 100%;

30 min, 0%; 32 min, 0%. The flow rate was 250 µl min$^{-1}$, the column temperature 50 °C, and the sample injection volume was 10 µl. The mass spectrometer operated at a resolution of 70,000 in polarity switching mode with the following electrospray ionization source conditions: spray voltage 3.5 kV; capillary temperature 300 °C; sheath gas 34; Aux gas 13; sweep gas 1 and probe temp 120 °C.

The occupancy of MQ9(II-H$_2$) associated with Huc was calculated by comparison against a single sample containing the standard, MQ9. MQ9 was the closest structural analogue to MQ9(II-H$_2$) (differing only in the presence of an additional double bond in the second isoprenyl unit) that was commercially available and was purchased from Toronto Research Chemicals. The peak area for the ammonium adduct of MQ9 in the standard sample containing 452.64 ng of MQ9 was $7.40 \times 10^9$ and the peak area for the ammonium adduct of MQ9(II-H$_2$) from the protein sample was $6.77 \times 10^9$ (Supplementary Fig. 5). Assuming that the response of MQ9(II-H$_2$) and MQ9 standard are equivalent, this provides an estimate of the amount of MQ9(II-H$_2$) in the protein sample of 414 ng. The protein sample contained 57 µg of Huc which corresponds to 68.4 pmoles (MW of Huc is 833 kDa). As there are eight MQ9(II-H$_2$) per Huc complex at full occupancy, this corresponds to 547.2 pmoles or 430.4 ng (MW of MQ9(II-H$_2$) = 786.63 g mol$^{-1}$) of MQ9(II-H$_2$). Consequently, the occupancy of Huc in MQ9(II-H$_2$) is estimated to be 96.2% (414 ng / 430.4 ng). If MQ9(II-H$_2$) is bound outside the observed binding sites in the Huc structure, then the actual occupancy in these sites will be lower. However, given the high quality of the density of the Huc head group present, we expect that the MQ9(II-H$_2$) present at these sites accounts for the majority in the sample.

### Mass spectrometric identification of HucM

The 18-kDa band on SDS–PAGE originating from the Huc complex was excised and extracted from the gel matrix. Upon destaining, the proteins were reduced with tris(2-carboxyethyl)phosphine (Pierce), alkylated with iodoacetamide (Sigma), and digested with mass spectrometry grade trypsin (Promega). The extracted peptides were analysed by LC–MS/MS on an Ultimate 3000 RSLCnano System (Dionex) coupled to an Orbitrap Fusion Tribrid (ThermoFisher Scientific) mass spectrometer equipped with a nanospray source. The peptides were first loaded and desalted on an Acclaim PepMap trap column (0.1 mm id × 20 mm, 5 µm) and then separated on an Acclaim PepMap analytical column (75 µm id × 50 cm, 2 µm) over a 30 min linear gradient of 4–36% acetonitrile/0.1% formic acid. The Orbitrap Fusion Tribrid was operated in data-dependent acquisition mode with a fixed cycle time of 2 s. The Orbitrap full ms1 scan was set to survey a mass range of 375–1,800 $m/z$ with a resolution of 120,000 at $m/z$ 400, an AGC target of $1 \times 10^6$, and a maximum injection time of 110 ms. Individual precursor ions were selected for HCD fragmentation (collision energy 32%) and subsequent fragment ms2 scan were acquired in the Orbitrap using a resolution of 60,000 at $m/z$ 400, an AGC target of $5 \times 10^5$, and a maximum injection time of 118 ms. The dynamic exclusion was set at ±10 ppm for 10 s after one occurrence. Raw data were processed using Byonic (ProteinMetrics v4.3) against a protein database covering *M. smegmatis* mc$^2$155. The precursor mass tolerance was set at 20 ppm, and fragment ions were detected at 0.6 Da. Oxidation (*M*) was set as dynamic modification, carbamidomethyl (*C*) as fixed modification. Only peptides and proteins falling below a false discovery rate of 0.01 were reported.

### Gas chromatography analysis

Gas chromatography experiments were used to measure the consumption of H$_2$ gas by both pure Huc protein and cultures of *M. smegmatis* expressing Huc. For pure protein experiments, to assess the rate of H$_2$ oxidation of Huc with menadione as the electron acceptor, 5 ml of buffer (50 mM Tris, 150 mM NaCl, pH 8.0, 0.3 mg ml$^{-1}$ BSA) with 200 µM menadione was contained in a 120 ml sealed serum vial, in triplicate. The headspace of the vial was flushed with 3 ppm H$_2$ in an atmospheric gas mix for 10 mins and then 100 nM Huc was added via a syringe to

begin the reaction. For experiments comparing the H$_2$ oxidation rate of Huc and Hyd1, the same reaction buffer contained 200 µM NBT in place of menadione and a headspace of either 10 or 100 ppm H$_2$ in N$_2$ was used, with 50 nM Huc or Hyd1 added to start the reaction. Before the experiment, Hyd1 was activated by incubation in an atmosphere of 7% H$_2$ in N$_2$ for 18 h, before being added anaerobically to the serum vials. Vials were incubated at room temperature, stirring, and hydrogen gas concentration was monitored using a pulsed discharge helium ionization detector (model TGA-6791-W-4U-2, Valco Instruments Company) over specific time intervals, until significant H$_2$ oxidation was no longer observed, or the limit of detection had been reached.

### Differential scanning fluorimetry

To determine the stability of the Huc complex, thermal melting was performed from 20 to 90 °C using a PrometheusNT.48 DSF (Nanotemper) using high sensitivity capillaries. The ratio of change in fluorescence at 330 and 350 nm was monitored to determine protein unfolding. Melting was performed with a Huc concentration of 0.2 mg ml$^{-1}$ in buffer containing 50 mM Tris, 150 mM NaCl pH 8.0.

### Cryo-EM imaging

Samples (3 µl) were applied onto a glow-discharged UltrAuFoil grid (Quantifoil GmbH) and were flash-frozen in liquid ethane using the Vitrobot mark IV (ThermoFisher Scientific) set at 100% humidity and 4 °C for the prep chamber. Data were collected on a Titan Krios microscope (ThermoFisher Scientific) operated at an accelerating voltage of 300 kV with a 50 µm C2 aperture at an indicated magnification of 105 kx in nanoprobe EFTEM mode. Gatan K3 direct electron detector positioned post a Gatan Quantum energy filter, operated in a zero-energy-loss mode with a slit width of 10 eV was used to acquire dose fractionated images of the Huc complex without an objective aperture. Two initial Huc datasets were collected, one low concentration dataset (0.4 mg ml$^{-1}$) composed of 2,226 movies, and a medium concentration dataset (1 mg ml$^{-1}$) composed of 3,113 movies. Movies were recorded in hardware-binned mode (previously called counted mode on the K3 camera) yielding a physical pixel size of 0.82 Å pixel$^{-1}$ with an exposure time of 6 s amounting to a total dose of 66.0 e$^-$ Å$^{-2}$ at a dose rate of -7.5 e$^-$ pixel$^{-1}$ s$^{-1}$, which was fractionated into 60 subframes. A further high concentration dataset (4 mg ml$^{-1}$) of 9,868 micrographs was also recorded using the same microscope but at a higher magnification of 165 kx on K3 detector, the resulting pixel size was 0.5 Å. The slit width for zero-loss filtering was set to 10 eV and images were collected at a dose rate of 4.37 e$^-$ pixel s$^{-1}$. Movies were collected with an exposure time of 4 s amounting to a total dose of 60.4 e$^-$ Å$^{-2}$, which were further fractionated into 60 subframes. The defocus range was set between −1.3 and −0.3 µm.

### Cryo-EM data processing and analysis

Micrographs from all datasets were motion-corrected using UCSF Motioncor 1.0.4 and dose-weighted averages had their contrast transfer function (CTF) parameters estimated using CTFFIND 4.1.8, implemented using Relion 3.1.2 (ref. [52]).

**Huc oligomer reconstruction from dataset 1.** Particle coordinates were determined by crYOLO 1.7.6 using a model trained on manually particles picked from 20 micrographs[53]. Unbinned particles were extracted from micrographs using Relion 3.1.2, before being imported into cryoSPARC 3.3.1 for initial 2D classification to remove bad particles, followed by ab initio model generation and 3D refinement[54]. Refined particles were reimported into Relion 3.1.2 and CTF refinement was performed, followed by Bayesian polishing[52]. Particles were reimported into cryoSPARC 3.3.1 for final 2D classification to remove residual bad particles, followed by non-uniform 3D consensus refinement to generate final maps at 2.19 Å (Fourier shell correlation (FSC) = 0.143, gold standard).

**HucS₂L₂ subunit dataset 1.** $HucS_2L_2$ particles were picked using gautomatch V 0.53 (developed by K. Zhang) using a diameter of 80 Å and a minimum distance between picked particles set to 20 Å. The resultant particles were extracted and binned 4 times to a box size of 56 pixels. A small subset of unbinned particles with 224 pixel box size were subjected to 2D classification to exclude noisy particles and the remainder were selected and subjected to 3D ab initio model generation with 4 classes using cryoSPARC v3.0.1 (ref. [54]). A class corresponding to a population of $HucS_2L_2$ subunits showed clear $C_2$ symmetry and this particle set was subjected to homogeneous refinement with $C_2$ symmetry applied resulting in a 3.25 Å resolution reconstruction. The resulting volume was used as the initial model for further classification and refinement of the full dataset as shown in the workflow Supplementary Fig. 3. Particles corresponding to the subunits were selected from binned dataset following 2D classification and heterogeneous refinement. The cryoSPARC 3D alignments from heterogeneous refinement were then exported to Relion using the csparc2star.py script in the pyem v0.5 package. In Relion, particles were re-extracted with no binning using the refined coordinates from the imported particles. These were re-imported into cryoSPARC and subjected to heterogeneous refinement followed by homogeneous refinement with $C_2$ symmetry applied. The FSC from the resultant reconstruction showed particle duplication as a result of over-picking using gautomatch. The refined particle coordinates were imported into Relion as described above and were cleaned to remove duplicated particles. The cleaned re-extracted particles were subjected to a round of CTF refinement followed by auto-refinement and Bayesian polishing. The resultant polished particles were imported back into cryoSPARC and CTF refinement, and non-uniform refinement with CTF parameter optimization was performed to reach a final global resolution of 1.67 Å (FSC = 0.143, gold standard), which was close to the Nyquist limit for the dataset (1.64 Å).

**HucS₂L₂ subunit dataset 2.** A second dataset at higher magnification was collected to overcome the Nyquist limit imposed on the map resolution of dataset 1. The particles were picked, extracted, and binned as described above and in Supplementary Fig. 2. Multiple rounds of 2D classification, ab initio classification, and heterogeneous refinement was performed to retain only particles containing the full Huc oligomer. Homogeneous refinement was performed on the binned particles with the $HucS_2L_2$ subunit map as an initial model in order to centre the particles on the $HucS_2L_2$ subunit. These particles were then re-extracted in Relion based on the refined coordinates to a box size of 380 pixels. Homogeneous refinement with $C_2$ symmetry resulted in a subunit reconstruction map of 2.15 Å. This step ensured the retention of all $HucS_2L_2$ particles with a good signal to noise ratio. In order to ensure that all symmetry-related $HucS_2L_2$ particles corresponding to the Huc oligomer are retained, the above-selected particles were re-extracted with a larger box size of 512 pixels and rescaled to 128 pixels (4× binned). The resultant particle set was subjected ab initio model generation followed by homogeneous refinement using $C_4$ symmetry to yield a 4.10 Å map of the Huc oligomer. The particles were re-extracted based on the refined coordinates from the previous step at a box size of 576 pixels and then rescaled to 288 pixels (2× binned). These particles were then subjected to homogeneous refinement followed by CTF refinement and Non-Uniform refinement in cryoSPARC to yield a 2.05 Å resolution map[54]. The refined particle set was then re-imported into Relion, and Bayesian polishing was performed. The resultant 'shiny' particles were extracted to a final box size of 576 pixels with no binning and were re-imported into cryoSPARC to perform symmetry expansion and final refinement. Duplicate particles were excluded to retain 153,359 Huc oligomers. Resultant particles were subjected to one round of refinement imposing $C_4$ symmetry, which yielded a 2.19 Å map. The resultant refined particles were symmetry expanded and subjected to local refinement within cryoSPARC 3.3.2 with no symmetry to yield a 2.16 Å map for $HucS_2L_2$. The resultant vol and particles were aligned to $C_2$ symmetry axis to exploit the $C_2$ symmetry within the $HucS_2L_2$. The symmetry aligned particles were then refined with $C_2$ symmetry followed by iterative Global CTF refinement, per particle defocus refinement followed by Ewald Sphere correction and a final round of per particle defocus refinement to reach a final global resolution of 1.52 Å (FSC = 0.143, gold standard).

**HucM stalk reconstruction.** The particle set resulting in a 2.15 Å map during the $HucS_2L_2$ subunit reconstruction from dataset 2 was re-extracted at a box size of 512 pixels and rescaled to 128 pixels (4× binned). The resultant particle set was subjected to 2D classification followed by ab initio model generation and homogeneous refinement using $C_4$ symmetry to yield a 4.10 Å map of Huc oligomer. The particles were re-extracted based on the refined coordinates from the previous step at a box size of 576 pixels and then rescaled to 288 pixels (2× binned). These particles were then subjected to homogeneous refinement followed by CTF refinement and non-uniform refinement in cryoSPARC to yield 2.05 Å resolution map[54]. The refined particles set was then re-imported into Relion and Bayesian polishing was performed, the resultant 'shiny' particles were extracted to a final box size of 576 pixels with no binning. These particles were subjected to ab initio model generation with three classes to retain particles that showed clear density for the HucM tail region. The selected particles were then ctf refined and Non-Uniform refined to yield a 2.09 Å map. The stalk density was isolated using volume tools in ChimeraX 1.3 and low pass filtered to 20 Å to derive the initial model for local refinement in cryosparc[55]. A dilated cosine padded softmask was generated using the initial model. The coordinates at the attachment of the stalk to the main body of the subunit were determined using the volume tracer tool in Chimera and used to define new fulcrum point for localized refinement (Supplementary Fig. 4). Two rounds of masked local refinement resulted in improving the density of the stalk region associated with the membrane, as shown in the figure to a resolution of 5.7 Å (no mask). The resultant FSC calculates a map resolution of 5.21 Å (FSC = 0.143, gold standard) but the map features are more consistent with a resolution of 6–8 Å on visual inspection.

## Huc model building and visualization

A model of the HucS and HucL subunit dimers was generated using Alpha-Fold and docked into one-half of the high-resolution Huc Dimer maps using ChimeraX 1.3 (refs. [56,57]). The model was refined and rebuilt into map density using Coot 0.92 (ref. [58]). [NiFe], [3Fe–4S], and menaquinone cofactors associated with Huc were downloaded from the PDB and customized restraints generated using Elbow within the PHENIX package before they were fitted and refined into maps using Coot[59,60]. The model was then refined using real-space refinement within PHENIX[61]. Once model building was complete the model was symmetry expanded using the Map symmetry tool and waters were added using DOUSE within the PHENIX package[60]. The refined dimer model was docked into the Huc oligomer map using ChimeraX[56], with HucM manually built into the available density for one subunit using Coot[58], followed by iterative real-space refinement within PHENIX and further model building. Once model building was complete the model was symmetry expanded using the Map symmetry tool within the PHENIX package[60]. Model quality was validated using MolProbity[62]. Images and movies were generated in Coot, ChimeraX, and Pymol[56,58]. The location and diameter of the gas channels of Huc and other [NiFe] hydrogenases was determined using the CAVER code[63].

## AlphaFold structural modelling

AlphaFold modelling was performed using AlphaFold version 2.1.1 implemented on the MASSIVE M3 computing cluster[57,64]. The sequence of the HucM C-terminal region (amino acids 80-189) was provided and modelling was run in multimer mode, with four molecules of HucM requested. The five ranked models produced by AlphaFold were compared for consistency with the top-ranked model used for further analysis and figure generation.

## Molecular dynamics parameterization and simulation

The 1.52 Å cryo-EM structure of Huc hydrogenase was used to construct the simulation system using the Charmm36m forcefield[65,66] with the TIP3P[67] water model. Sodium counterions were added to the solution to neutralize the system charge. Three sets of simulations were set up containing no gas molecules or containing 250 hydrogen or 250 oxygen molecules enriched in solution. Gas molecules were randomly inserted into the water phase. Each simulation was initiated with different starting velocities and gas positions and three independent simulations of each system were performed. Simulations were run for 50 ns (wild-type) or 100 ns (mutants). Molecular visualization and analyses were performed using VMD software[68]. The topologies of the metal cofactors were constructed based on the coordinates from the cryo-EM structure. Previous work on hydrogenase cofactors[69] has demonstrated the transferability of structurally identical metallic cofactors between hydrogenases. Thus, atomic partial charges and force constants for the bond lengths of the metal cofactors were taken from previous studies of the structurally similar *Desulfovibrio fructosovorans* [NiFe] hydrogenase[69]. Equilibrium values of bond lengths and angles within the metal cofactor were taken directly from the cryo-EM structure. Molecular dynamics simulations were performed in a single redox state. The ready active Ni-SI form of the active site and oxidized form of the iron–sulfur clusters were chosen based on previous similar analyses[70]. The [NiFe] cofactor had bond constraints placed across the CN and CO ligands as well as between the iron atom and two bound sulfur atoms. Lennard–Jones parameters were taken from the Charmm36m[65,66] forcefield where appropriate, with exception of the cyanuric carbon[71] with the iron parameters taken from physical data to have a non-zero epsilon value[72]. The QL molecular oxygen model developed by Javanainen et al.[73] was chosen to better define the quadrupole within the molecule. The oxygen atoms are given mass and non-zero Lennard–Jones parameters as well as a partial charge. A massless virtual site, OD1, exists at the midpoint of two oxygen atoms to carry the neutralizing charge, simulating the charge distribution within the oxygen quadrupole. Likewise, the hydrogen model was given mass and charge distribution coherent with the work of Hunter et al.[70]. Molecular dynamics simulations were performed with the GROMACS v 2021.3 simulations suite. Energy minimization was performed using the steepest descent algorithm followed by a second minimization step using the conjugate gradient approach, both set to a maximum force of 250 kJ mol$^{-1}$ nm$^{-1}$. Equilibration simulations were performed within the NVT ensemble at 310 K. Position restraints were applied to the protein, cofactors and magnesium ion, with a 10$^3$ kJ mol$^{-1}$ nm$^{-2}$ force constant for 100 ps. Further equilibration was carried out within the NPT ensemble with pressure maintained at 1 bar using the Parrinello–Rahman barostat[74], for 100 ps. Production simulations in the NPT ensemble were each of duration 50 ns, at a temperature of 310 K. Long-range electrostatics were treated with the Particle Mesh Ewald method[75], with values for the real-space long-range cut-off for these and Van der Waals forces being set to 10 Å. Covalent bonds containing hydrogen atoms were restrained using the LINCS algorithm[76,77] allowing a 2 fs integration step.

## EPR spectroscopy

The composition of the iron–sulfur clusters of Huc was analysed by EPR spectroscopy. X-band EPR measurements were performed on a Bruker ELEXYS E500 spectrometer equipped with a SuperX EPR049 microwave bridge and a cylindrical TE$_{011}$ ER 4122SHQE cavity in connection with an Oxford Instruments continuous flow cryostat. Measuring temperatures were achieved using liquid helium flow through an ITC 503 temperature controller (Oxford Instruments). Samples of Huc isolated in air were prepared under a neat argon atmosphere or flash-frozen following incubation under 1 atm H$_2$, and X-band EPR spectra were collected in the temperature range 40–7 K, at varying microwave powers. Preliminary simulations of the Ni signals in the EPR spectra were carried out using EasySpin v.6.0.0-dev47 (ref. [78]).

## FTIR spectroscopy

A 4 μl volume of 5 mg ml$^{-1}$ Huc enzyme solution in a buffer containing 50 mM Tris, 150 mM NaCl pH 8 was deposited on an ATR crystal surface. The sample was applied under laboratory atmosphere (air), dried under 100% nitrogen gas, and rehydrated with a humidified aerosol (100 mM Tris-HCl (pH 8)) as described previously[79,80]. A custom-built PEEK cell (inspired by Stripp et al.[81]) that allows for gas exchange sealed the ATR unit (BioRadII from Harrick) mounted in the FTIR spectrometer (Vertex V70v, Bruker). Spectra were recorded with 2 cm$^{-1}$ resolution, 80 Hz scanner velocity, and averaged over a varying number of scans (at least 100 scans). All experiments were performed at ambient conditions (room temperature and pressure, hydrated enzyme films). Data were analysed using OPUS 8 and Origin 2021 software. To assess the response of the Huc [NiFe] cluster to ambient air, N$_2$, and H$_2$, the PEEK cell was flushed sequentially with each gas in the following sequence: N$_2$ (ambient air isolated enzyme), $^2$H$_2$, N$_2$, H$_2$, ambient air. Changes in the Huc spectra were allowed to stabilize before the spectra in each gas were collected.

## Protein film electrochemistry

**Huc in 100% H$_2$ experiment.** Protein film electrochemistry experiments were carried out under anaerobic conditions. The 3-electrode system was made up of (1) Ag/AgCl (4 M KCl) as reference electrode, (2) 5 mm diameter pyrolytic graphite edge (PGE) encapsulated in epoxy as the rotating disk working electrode, and (3) graphite rod as the counter electrode. The glass cell featured a water jacket for temperature control and a gas inlet/outlet for H$_2$ flow. The buffer used was composed of 5 mM 2-[N′-morpholino]-ethane sulfonic acid (MES), 5 mM 2-[N′-cyclohexylamino]ethane sulfonic acid (CHES), 5 mM N′-[2-hydroxyethyl]piperazine-N′-2- ethane sulfonic acid (HEPES), 5 mM N′-tris[hydroxymethyl]methyl-3-amino propane sulfonic acid (TAPS), 5 mM sodium acetate, with 0.1 M Na$_2$SO$_4$ as carrying electrolyte, titrated with H$_2$SO$_4$ to pH 7.0 at 20 °C, and purged with N$_2$ for 3 to 4 h. To remove residual O$_2$ in the PGE electrode, cyclic voltammograms were run at 100 mV s$^{-1}$ from 0 to −800 mV (versus SHE, standard hydrogen electrode) for 10 scans. To adhere Huc to the deaerated PGE electrode the surface was abraded with P1200 sandpaper and rinsed with purified water. An aliquot of 5 mg ml$^{-1}$ Huc enzyme solution in a buffer containing 50 mM Tris, 150 mM NaCl pH 8 was mixed with polymyxin B sulfate and transferred to the electrode surface. The cell solution was then saturated with H$_2$ (1 bar, 10 min) before the cyclic voltammogram of the system with the immobilized enzyme was recorded at 10 mV s$^{-1}$. The cyclic voltammogram of the blank electrode was recorded with no enzyme immobilized. Electrochemical data were acquired using a PGSTAT10 and GPES 4.9 software (Metrohm/Autolab). Data were analysed using Origin 8 software. All potential values are referenced versus SHE. Experiments were conducted on two independent Huc films, with each film scanned at least two times.

**Huc, Hyd1 and Hyd2 in 0 to 5% H$_2$ experiments.** Protein film electrochemistry experiments were conducted in a N$_2$-filled anaerobic glove box (Glove Box Technology Limited, O$_2$ < 2 ppm). Experiments were carried out using a glass electrochemical cell with gas inlet and outlet valve[82]. The main cell compartment was surrounded by a water jacket to control temperature. All experiments were performed at 25 °C. A saturated calomel electrode (SCE) was used as the reference, and was held in an isolated glass arm containing 0.10 M NaCl and connected to the main cell compartment by a Luggin capillary. The SCE was calibrated with ferrocene-methanol as +244 mV versus SHE at 25 °C, and potentials are corrected back to V versus SHE in all figures. The working electrode was a PGE rotating disk electrode, 2 mm diameter, and was polished with silicon carbide paper (first P1200 and then P4000). Electrode rotation was controlled by a Metrohm Autolab IME663 rotator, and the electrode was rotated at 2,000 rpm in all experiments shown here in order to achieve effective mass transport to/from the immobilized enzyme

film. A platinum wire was used as the counter electrode. Cyclic voltammetry was controlled by an Autolab PGSTAT 10 potentiostat using NOVA software version 1.10. All experiments were conducted with gas flushing through the headspace of the electrochemical cell. Gas mixtures were prepared from cylinders of pure Ar, pure $H_2$, 5% $H_2$ in $N_2$, 1,000 ppm $H_2$ in $N_2$ (all from BOC), using gas mass flow controllers (Smart-Trak2, Sierra Instruments) to obtain the desired partial pressure of $H_2$ within an inert carrier gas (Ar or $N_2$). A mixed buffer system was used in all experiments; this consisted of 15 mM in each of MES, HEPES, TAPS, CHES (all from Melford) and sodium acetate (Sigma), with 0.1 M NaCl (Fisher) as supporting electrolyte. All solutions were prepared using purified water (Millipore: resistivity 18.2 MΩ cm) and titrated with HCl to pH 7.0 at 25 °C. Buffers were flushed with $N_2$ overnight to remove $O_2$ before being taken into the glove box. For preparation of films of Huc, 2 μl enzyme solution (containing 20 mg ml$^{-1}$ polymyxin B sulfate as coadsorbate) was spotted onto a freshly polished PGE electrode surface and left for 3 min; the electrode was then rinsed with purified water to remove unabsorbed enzyme. The electrode was placed in the electrochemical cell containing 8 ml of pristine mixed buffer. To minimize protein film loss during measurements, 0.025 mg ml$^{-1}$ polymyxin B sulfate was added to the mixed buffer. The cell headspace was flushed with gas for at least 15 min to fully equilibrate with the cell solution before applying a potential. Cyclic voltammograms were recorded with constant gas flow through the headspace of the cell at 1 bar. *E. coli* Hyd1 and Hyd2 were prepared as described previously[83,84]. These hydrogenases were first activated under 100% $H_2$ at room temperature for at least 5 h. Films of Hyd1 and Hyd2 were prepared as for Huc, but without use of polymyxin B sulfate.

### Reporting summary

Further information on research design is available in the Nature Portfolio Reporting Summary linked to this article.

## Data availability

Cryo-EM maps and atomic models generated in this paper have been deposited in the Protein Data Bank (accession codes 7UTD, 7UUR, 7UUS, 8DQV) and the Electron Microscopy Data Bank (accession codes EMD-26767, EMD-26801, EMD-26802, EMD-27661). Raw data from the Huc molecular dynamics simulations are available via Zenodo (https://zenodo.org/record/7378976). Source data are provided with this paper.

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

**Acknowledgements** We thank the Bio21 Institute for the use of the Thermo Glacios housed in their facilities during cryo-EM sample preparation and screening. We thank S. Carr for providing purified Hyd1 and Hyd2; S. Frielingsdorf, J. Rossjohn, F. Armstrong, B. Murphy and G. Knott for their review and constructive feedback on the manuscript. We acknowledge the use of instruments and assistance at the Monash Ramaciotti Centre for Cryo-Electron Microscopy, a Node of Microscopy Australia. This work was supported by an Australian Research Council (ARC) DECRA Fellowship (DE170100310) (to C.G.), an ARC Discovery Project Grant (DP200103074) (to R.G. and C.G.), a National Health and Medical Research Council Emerging Leader Grant (NHMRC) (APP1178715) (to C.G.), a NHMRC Emerging Leader Grant (APP1197376) (to R.G.), ARC LIEF grants (LE200100045, LE120100090) for the Titan Krios Gatan K3 Camera and for the Titan Krios, the Deutsche Forschungsgemeinschaft through Priority Program 1927 (1554/5-1) (to S.T.S.), a Swedish Energy Agency (grant no 48574-1) (to G.B.), a European Research Council grant (714102) (to G.B.), the European Union's Horizon 2020 research and innovation program (Marie Skłodowska Curie Grant no. 897555) (to M.S.), and a United States National Science Foundation grant (CHE-2108684) (to H.S.S.).

**Author contributions** C.G., R.G., A.K. and G.M.C. conceived and designed the project. C.G. and R.G. administered the project. A.K. and R.G. generated *M. smegmatis* mutants, purified proteins, performed kinetic analysis and $H_2$ consumption assays. H.V., R.G., A.K. and M.B. prepared cryo-EM grids and collected and processed cryo-EM data. R.G. modelled and analysed the Huc structures. G.B., K.A.V., P.R.C. and Z.D. devised, performed and analysed the PFE experiments. G.B., P.H. and H.S.S. devised, performed and analysed the EPR experiments. G.B., M.S. and S.T.S. devised, performed and analysed the FTIR experiments. S.K., J.B., R.J. and R.G. designed, performed and analysed the molecular dynamics simulations. R.B.S. and C.K.B. designed and performed the mass spectrometry experiments. R.G., C.G. and A.K. wrote and edited the manuscript with input from all authors.

**Competing interests** The authors declare no competing interests.

**Additional information**
**Correspondence and requests for materials** should be addressed to Rhys Grinter or Chris Greening.

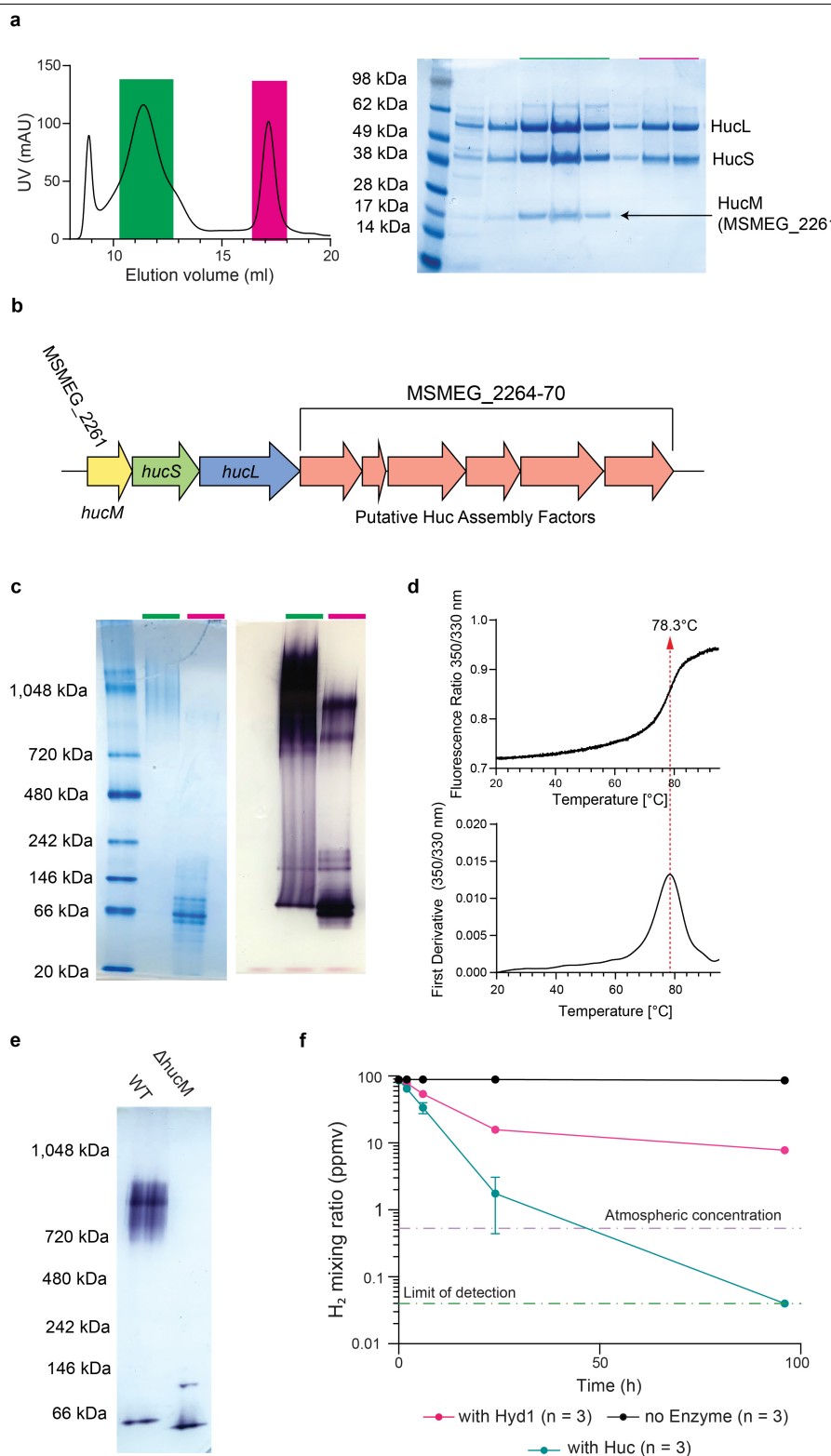

**Extended Data Fig. 1 | Huc isolation and purification.** (a) Left panel: A chromatogram showing streptactin-purified Huc separated via size exclusion chromatography on a Superose 6 10/300 column. The green highlighted region contains the Huc oligomer, while the pink region contains a low molecular weight Huc species. Right panel: A Coomassie-stained SDS-PAGE gel showing fractions from the coloured peak regions of the chromatogram. (b) A schematic of the Huc gene cluster showing the location of *hucM* (MSMEG_2261) compared to *hucS* and *hucL*. (c) A native-PAGE gel of the purified Huc oligomer (green) and low molecular species (pink), stained with Coomassie (left panel)

and NBT (right panel). (d) Differential scanning fluorimetry analysis of the Huc oligomer showing a transition in tryptophan-derived fluorescence at 78.3 °C, indicating the melting temperature of the oligomer. (e) An NBT-stained native-PAGE gel of *M. smegmatis* cell lysate showing that Huc does not form an oligomer in the *ΔhucM* strain. (f) Gas chromatography analysis of the $H_2$ concentration of the headspace of sealed vials containing Huc, *E. coli* Hyd1 or no enzyme, showing Huc can oxidize $H_2$ from 100 ppm to the limit of detection for the gas chromatograph (40 ppbv) (green line). In contrast, Hyd1 does not oxidise $H_2 < 8$ ppmv. Data are presented as mean values +/− SD.

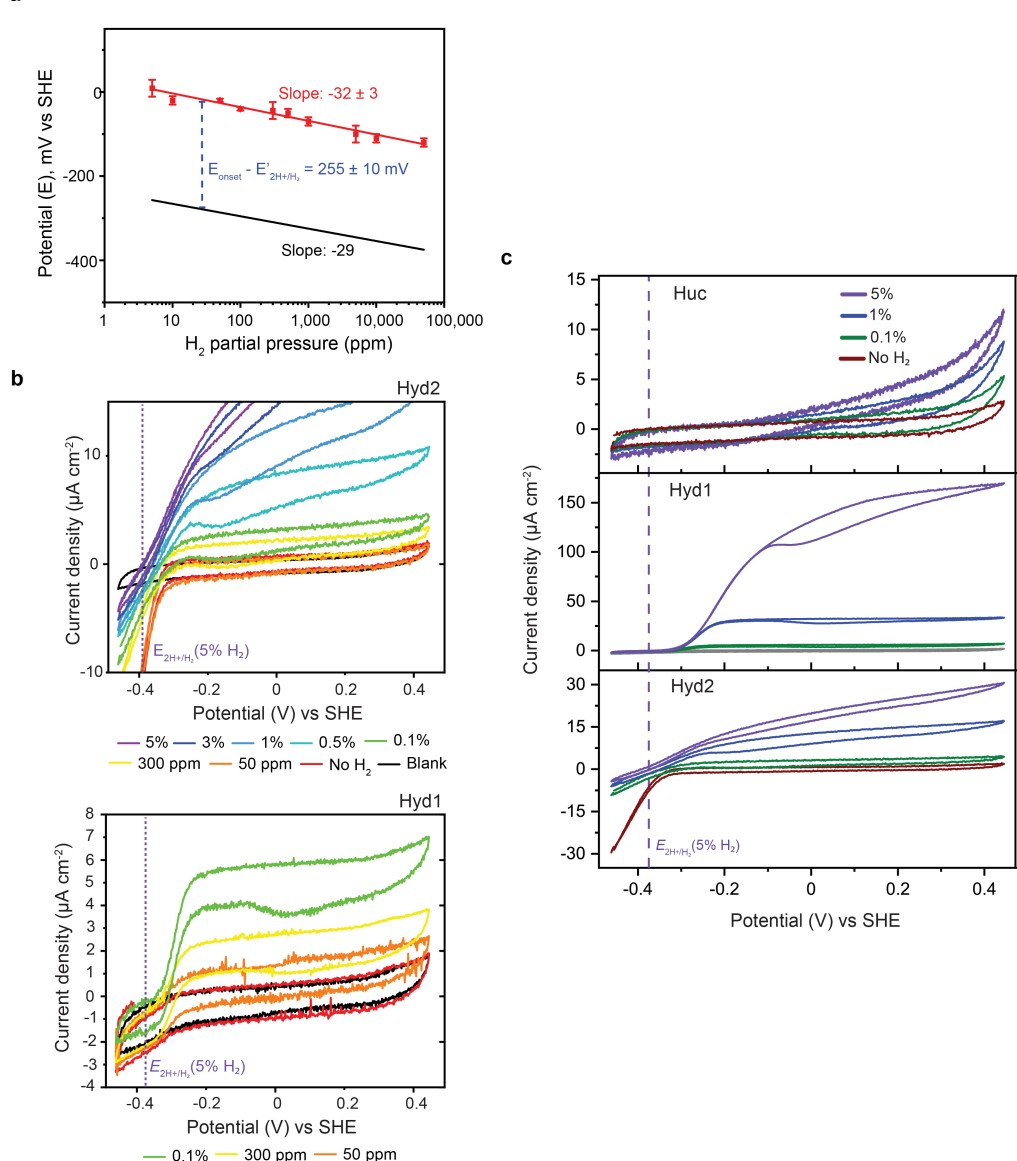

**Extended Data Fig. 2 | Comparison of protein film electrochemistry response of Huc with Hyd1 and Hyd2 from *E. coli* at different H₂ concentrations.** (a) The Huc onset potential calculated from data presented in Fig. 1f, and the calculated $E^{0'}_{2H^+/H_2}$ potential as a function of H₂ concentration, showing that the difference between these potentials (the overpotential, η) remains constant regardless of H₂ concentration. Data are presented as mean values +/− SD (b) The behavior of *E. coli* Hyd2 and Hyd1 at low H₂. Cyclic voltammograms were recorded with immobilized hydrogenase with a gas mixture flushing through the headspace comprising argon, N₂ and the indicated level of H₂. Top: *E. coli* Hyd2; Bottom: *E. coli* Hyd1. The dashed vertical line indicates the potential of the H⁺/H₂ couple, calculated for pH 7, 5% H₂ at ambient pressure. (c) Cyclic voltammograms with immobilized hydrogenase, in enzyme-free mixed buffer (pH 7.0, 25 °C) with a gas mixture

flushing through the headspace comprising argon, N₂ and the indicated level of H₂. The dashed vertical line indicates the potential of the H⁺/H₂ couple, calculated for pH 7, 5% H₂ at ambient pressure. For each enzyme, the set of scans was recorded on the same enzyme film in the order of low to high H₂. Top panel: Huc; middle panel: *E. coli* Hyd1; bottom panel: *E. coli* Hyd2. At 5% H₂, Hyd1 shows an onset potential of −340 mV for H₂ oxidation, more negative than the onset observed for Huc of −120 mV. In contrast, Hyd2 shows reversible catalysis at 5% H₂ with the current for H⁺ reduction and H₂ oxidation cutting the zero current potential very close to the thermodynamic $E_{2H^+/H_2}$ potential of −375 mV. The H⁺ reduction current for Hyd2 increases in magnitude as the H₂ concentration is lowered and product inhibition is relieved.

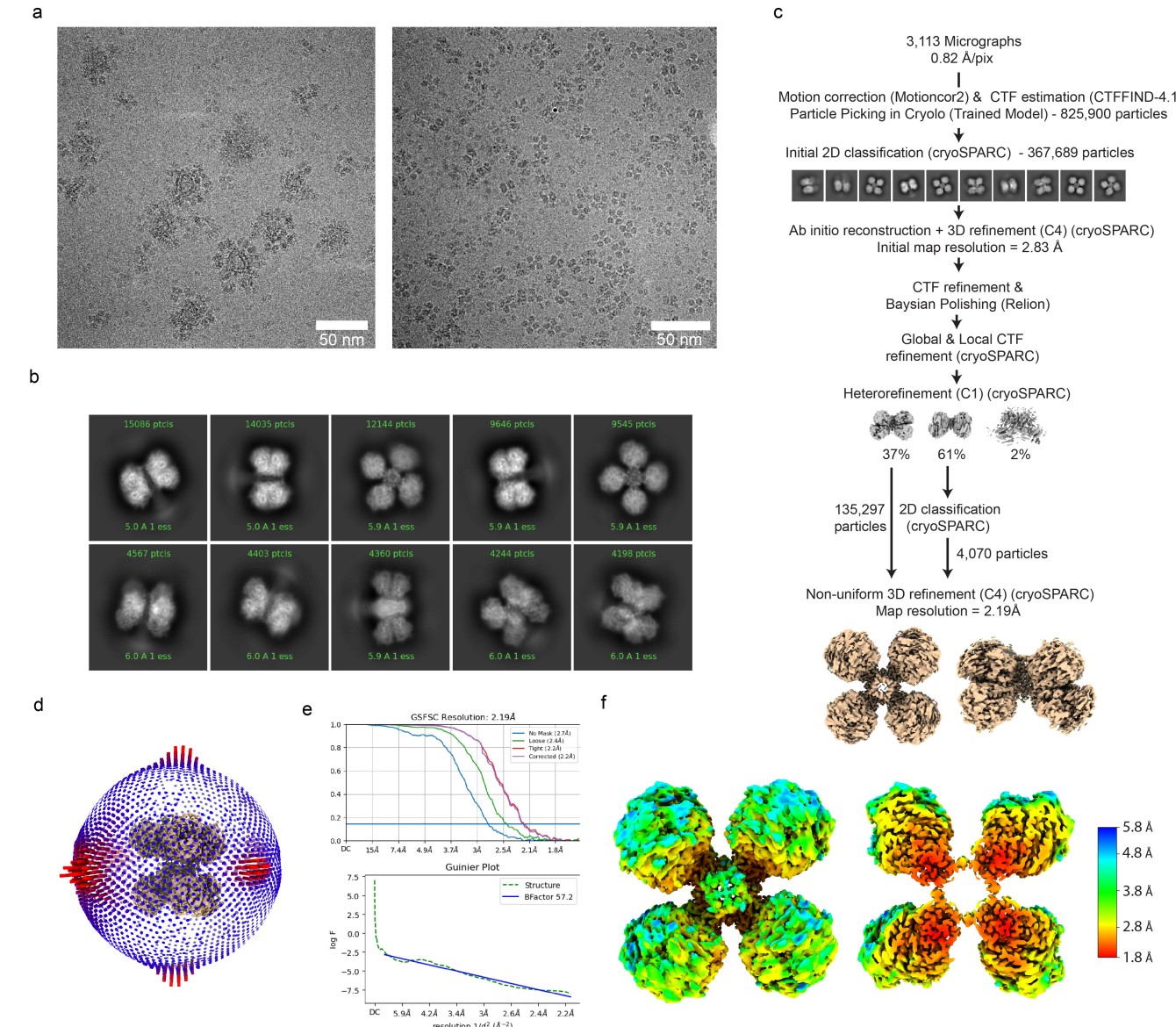

**Extended Data Fig. 3 | Cryo-EM visualisation and 3D reconstruction of the Huc oligomer.** (a) Motion-corrected micrographs of vitrified purified Huc oligomers, showing Huc associated with membrane vesicles (left panel) and free Huc (right panel). (b) Selected 2D class averages of Huc oligomer, showing C4 symmetry and flexible membrane-associated stalk. (c) Data processing workflow for the Huc oligomer reconstruction. (d) The Euler angle distribution of particles used for Huc oligomer reconstruction. (e) Gold-standard Fourier shell correlation (FSC) curves calculated from two independently refined half-maps indicate an overall resolution of 2.19 Å at FSC = 0.143, and Guinier plot indicates a sharpening B-factor of 57.2. (f) A local resolution map of the Huc oligomer, showing a resolution range of -1.8 to 4.8 Å from the Huc core to the periphery, indicative of significant interdomain motion.

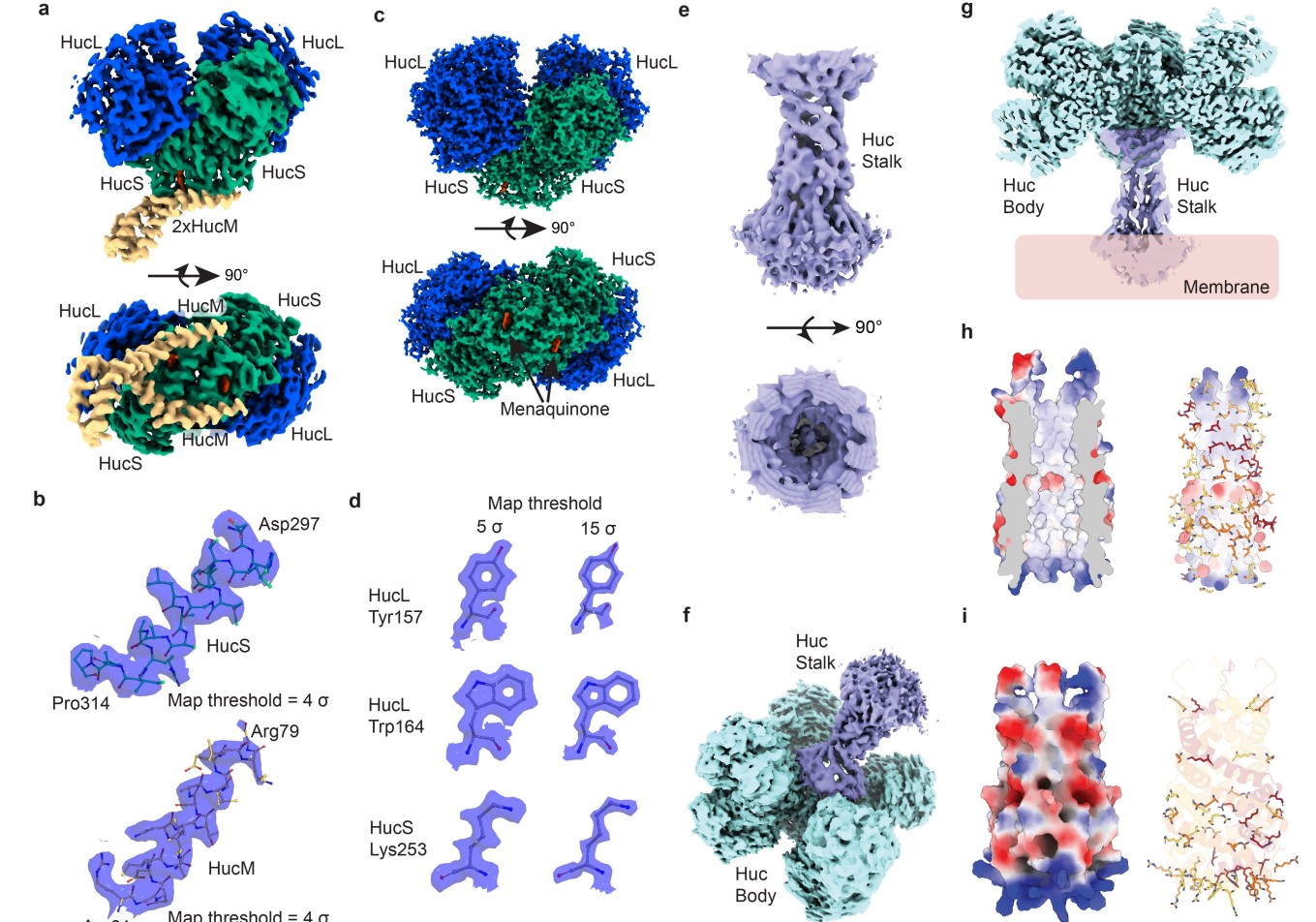

**Extended Data Fig. 4 | Map quality of Huc oligomer and dimer Cryo-EM reconstructions and Cryo-EM density map and AlphaFold model of the Huc stalk region.** (a) Maps from the 2.19 Å Huc oligomer reconstruction for one Huc lobe (HucS$_2$L$_2$) and 2 HucM molecules. (b) Helices from HucS and HucM with associated density from oligomer reconstruction. (c) Maps from the 1.52 Å Huc dimer reconstruction. (d) Examples of density corresponding to amino acids from the Huc dimer reconstruction contoured at 5 or 15 σ. (e) The stalk region from Cryo-EM density maps from Supplementary Fig. 4. (f) A composite of the Huc stalk and body map regions from the 2.19 Å dataset. (g) A cut-away view of the composite map from panel f, showing the enclosed chamber and the centre of the Huc complex. (h) A surface view of the AlphaFold model of the HucM C-terminal stalk region shown as a cutaway electrostatic surface (left) and the internal surface lined with hydrophobic residues (right). (i) The HucM C-terminal region as in panel d, showing the positively charged region at the base of the stalk (left) and the high frequency of arginine and lysine residues in this region (right).

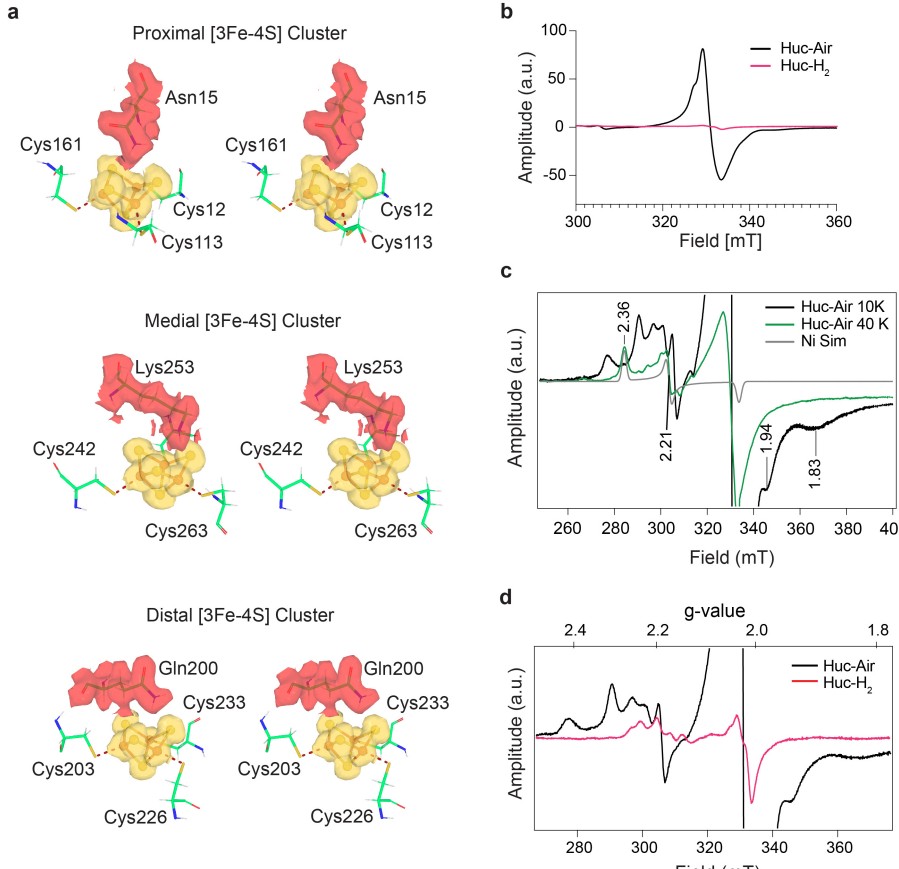

**Extended Data Fig. 5 | Assignment of HucS iron sulfur clusters as 3×[3Fe-4S] and the Ni-B state of oxidized Huc.** (a) Cross-eye stereo images of carved density maps corresponding to the three HucS [FeS]-clusters (yellow) and the additional non-cystine [FeS]-cluster interacting residues (red). Density maps are derived from the 1.52 Å Huc dimer reconstruction contoured to 5 σ. (b) EPR spectra collected from a Huc solution in the as-isolated oxidized state (Huc-Air) and $H_2$ reduced state (Huc-$H_2$) at low T (7 K); Modulation frequency 100 kHz, amplitude 10 G; Microwave frequency 9.4 GHz; power 2 mW. Spectra collected at low T of $H_2$-flushed samples showed no discernible contribution from rhombic components typically associated with reduced [4Fe-4S]$^+$ clusters. (c) Zoomed-in view of the EPR spectra of as-isolated Huc measured at 10 K (black) and 40 K (green). The presence of multiple peaks at low T is consistent with strong spin-spin interactions between the EPR-active Ni center and the [3Fe-4S]$^+$ cluster, giving rise to the "split" nickel signals observed. At higher temperatures, these split signals collapse, and a spectrum more characteristic of a canonical Ni-B signal is recovered. The simulation of a single species with g = 2.36, 2.22, and 2.00 is overlaid; we note that the third g-value is unresolved because of overlap with the cluster signal. Additional split features of the [3Fe-4S]$^+$ cluster are indicated at lower g-values. (d) Zoomed-in view comparing the EPR spectra of as-isolated (black) and $H_2$-reduced (red) Huc at 7 K. The nickel signals are seen to shift considerably upon $H_2$ incubation. The observed reduction in spectral anisotropy is consistent with the changes expected in converting from a Ni-B to a Ni-C species, though some splitting of the signals are still observed.

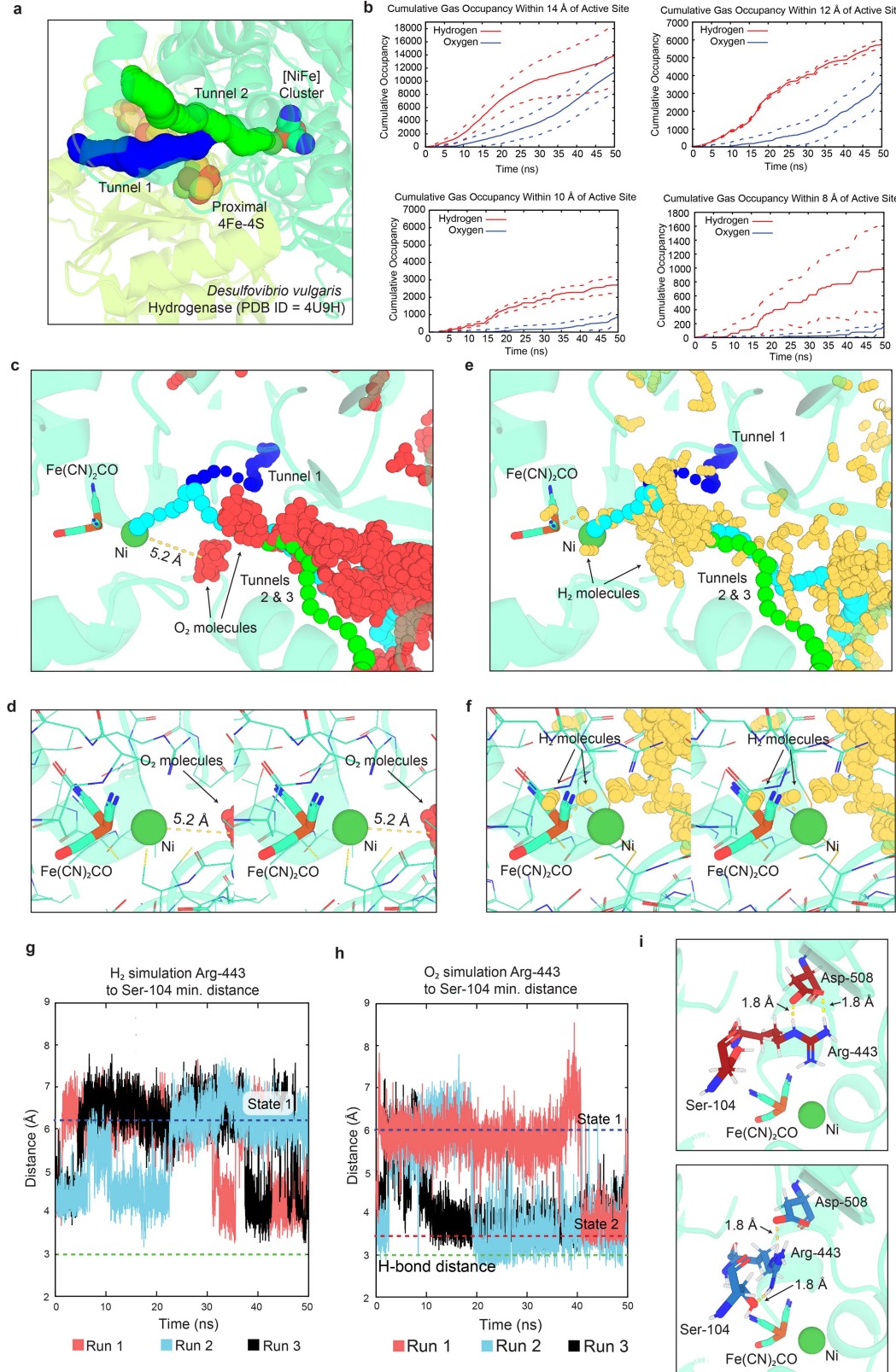

**Extended Data Fig. 6 | See next page for caption.**

**Extended Data Fig. 6 | Molecular dynamics simulations of HucSL in the presence of $H_2$ or $O_2$.** (a) A zoomed view of the large and small subunits of an [NiFe]-hydrogenase from *Desulfovibrio vulgaris* (PDB ID = 4U9H)[24], showing the location and width of the hydrophobic gas channels that provide substrate access to the [NiFe] active site. See Supplemental Table 2 for tunnel width statistics. (b) Cumulative occupancy plots showing the proximity of $H_2$ and $O_2$ molecules to the [NiFe] cluster of Huc throughout the molecular dynamics simulations. (c) An expanded view of the positions of a representative subset of $O_2$ molecules in closest proximity to the Huc [NiFe] cluster in molecular dynamics simulations, with the path of the hydrophobic gas channels shown as blue, green, and cyan spheres. (d) A zoomed stereo view of $O_2$ as described in panel c. (e) and (f) The positions of a representative subset of $H_2$ molecules in closest proximity to the Huc [NiFe] cluster are displayed as described in panels c and d. (g) A distance-over-time plot showing the relative proximity of serine 104 to arginine 443, in the Huc MD simulations in the presence of $H_2$. (h) A distance-over-time plot showing the relative proximity of serine 104 to arginine 443, in the Huc simulations in the presence of $O_2$. (i) The relative position of arginine 443 in states 1 (top panel) and 2 (bottom panel) populated during the Huc MD simulations.

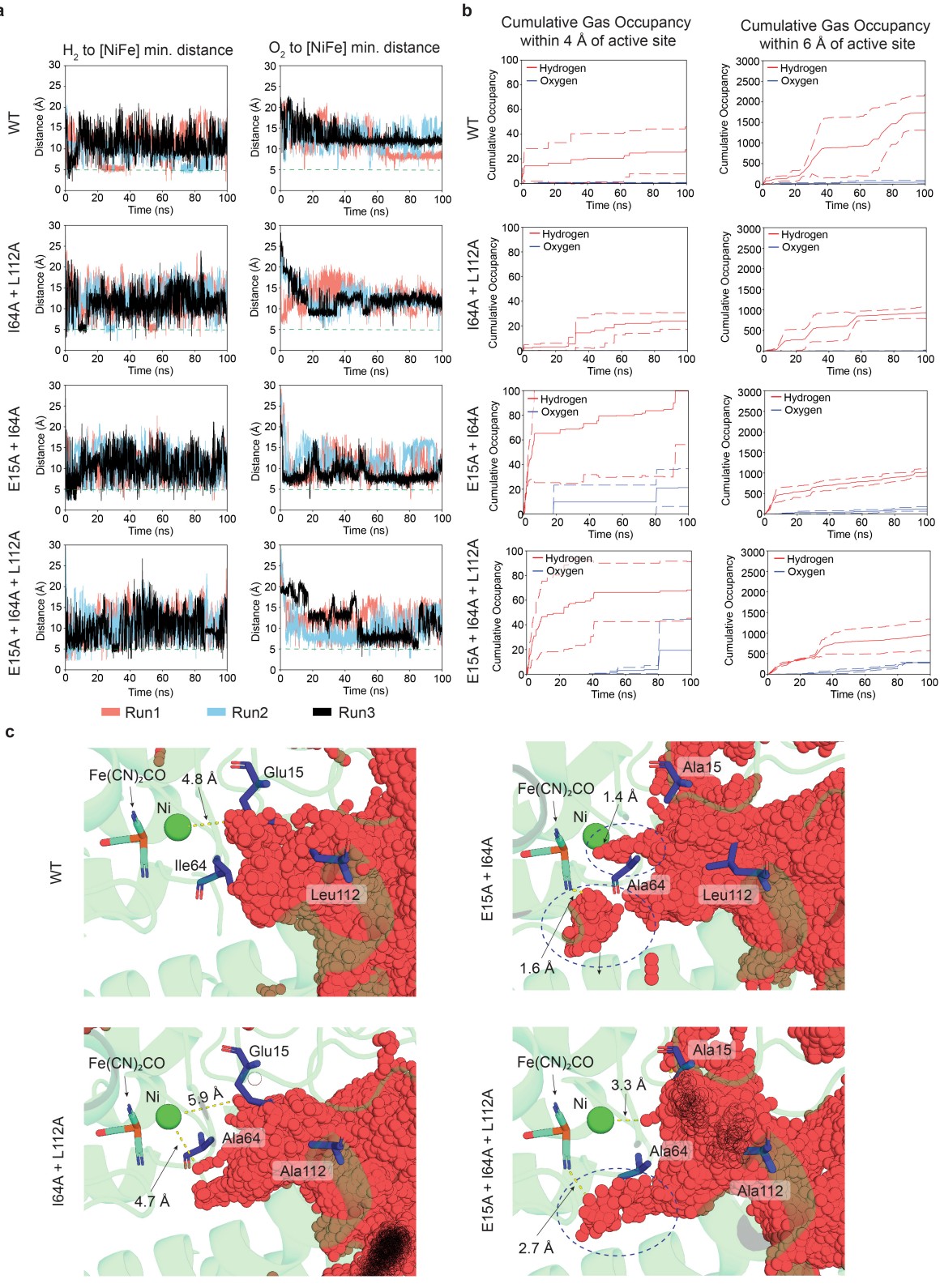

**Extended Data Fig. 7 | Molecular dynamics simulations showing oxygen accessibility to the catalytic site of Huc gas channel mutants.** (a) Distance-over-time plots showing the relative proximity of $H_2$ and $O_2$ to the [NiFe] cluster of WT and gas channel mutant Huc variants in the Huc MD simulations. (b) Cumulative occupancy plots showing the proximity of $H_2$ and $O_2$ molecules to the [NiFe]-cluster of WT and gas channel mutant Huc variants throughout the molecular dynamics simulations. (c) A structural representation of the region of Huc encompassing the catalytic cluster from the WT and gas channel mutant Huc variants from Run 1 of the MD simulations, showing the position of $O_2$ molecules in this region at each frame of the simulations. Residues targeted for mutagenesis are shown as sticks, and regions of enhanced $O_2$ access in the E15A + I64A and E15A + I64A + L112A mutants are indicated.

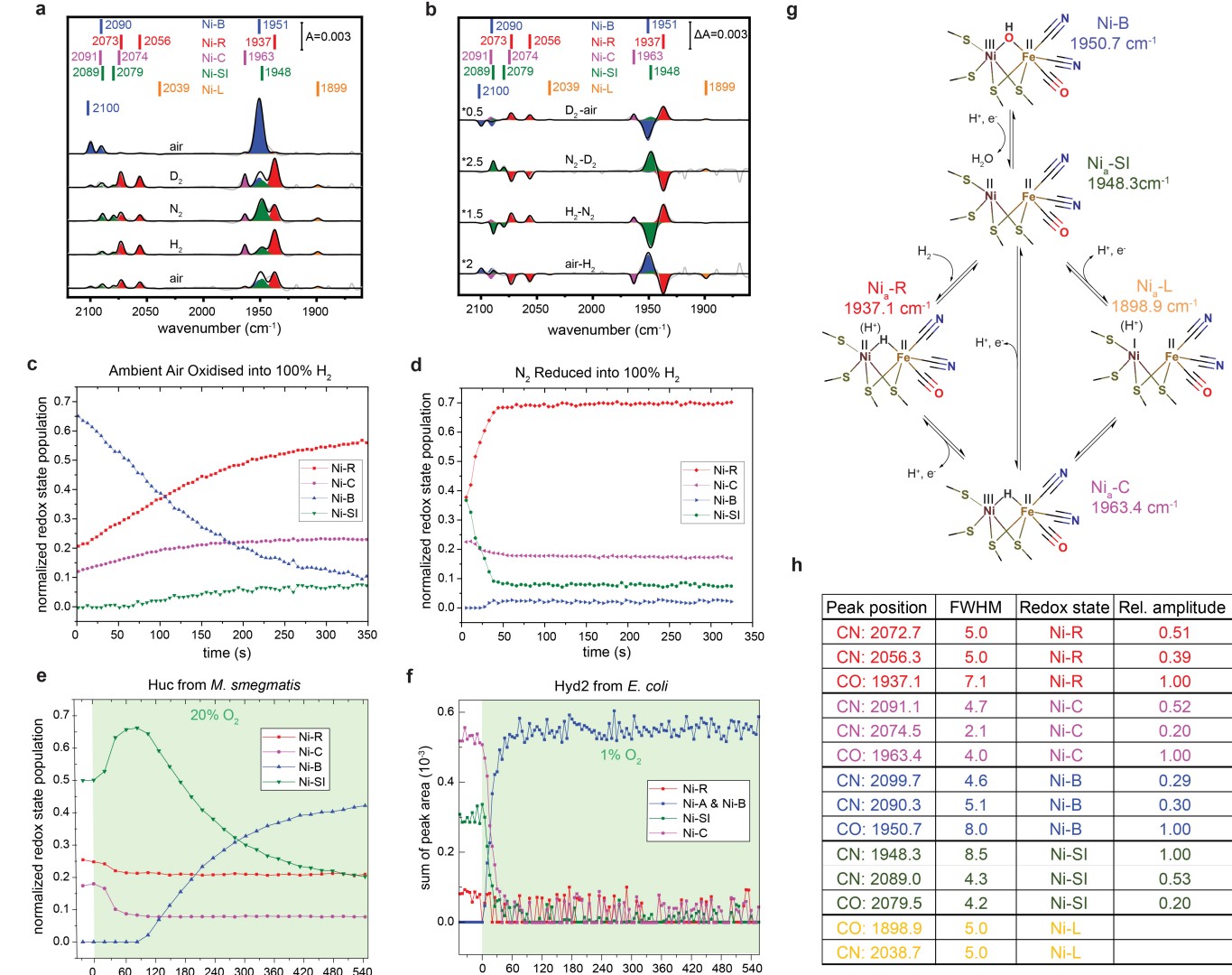

**Extended Data Fig. 8 | FTIR analysis of Huc in the presence of ambient air, N₂ or H₂.** (a) Absolute FTIR spectra of Huc equilibrated in air, followed by sequential equilibration in a 100% atmosphere of the labelled gas, followed by air. [NiFe] cluster states are assigned based on literature-derived values (see Supplementary Table 3). (b) Difference spectra derived from spectra shown in panel a. (c) A time-course plot showing the relative fractions of the FTIR assigned states immediately following the transfer of Huc from ambient air to a 100% H₂ atmosphere, the catalytic Ni-R state becomes populated at the expense of hydroxide bound Ni-B over a time scale of minutes. (d) A time-course plot showing the relative fractions of the FTIR assigned states immediately following the transfer of Huc in a mixed Ni-SI, Ni-R state from 100% N₂ to a 100% H₂ atmosphere. The catalytic Ni-R state becomes populated at the expense of the non-hydrogen bound reduced Ni-SI state over a timescale of seconds.

(e) A time-course plot of [NiFe] cluster states following the transfer of Huc predominantly in the Ni-SI state from 100% N₂ into 80%:20% N₂ to O₂. Initially, the Ni-SI state is populated at the expense of the hydrogen-bound Ni-C and Ni-R states, followed by population of the Ni-B state at the expense of Ni-SI over a timescale of minutes. (f) A time-course plot of [NiFe] cluster states of Hyd2 from *E. coli* following the transfer from 100% N₂ into 99%:1% N₂ to O₂. The Ni-B state is rapidly populated at the expense of all other states over a timescale of seconds. Data from Senger et al. 2018[85]. (g) A scheme of the proposed catalytic cycle for [NiFe]-hydrogenases, showing the states occupied by Huc during FTIR analysis. (h) Vibrational frequencies for the CO and CN bands identified from FTIR spectra for each of the proposed Huc states. Peak position and FWHM units = cm⁻¹.

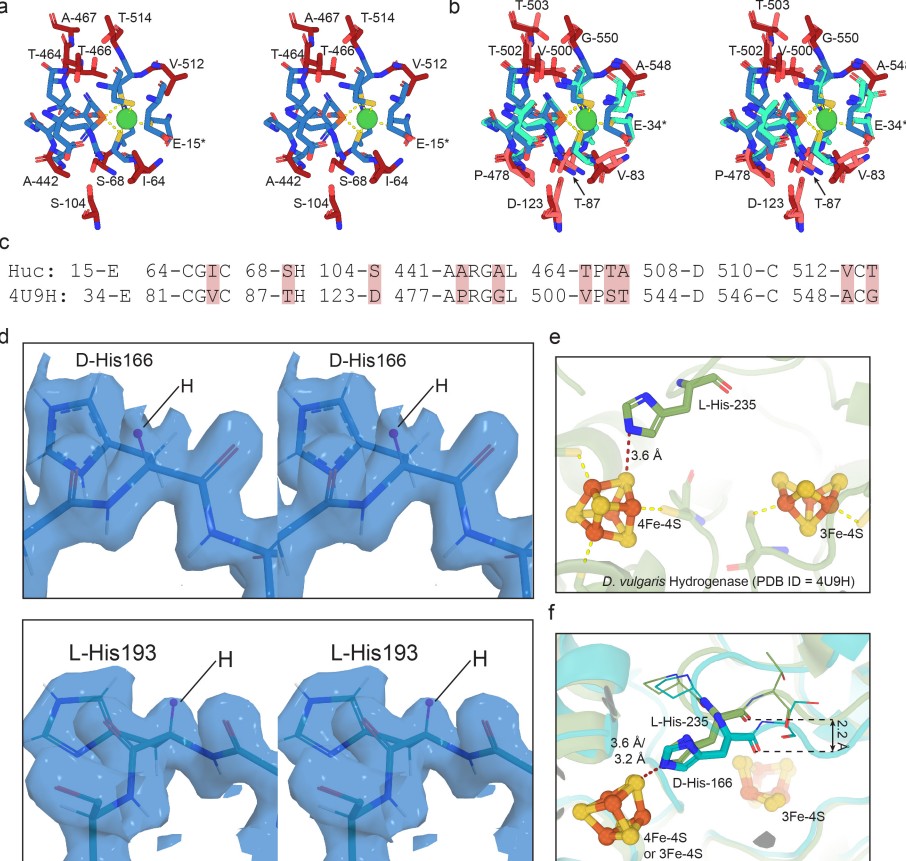

**c**

```
Huc:  15-E   64-CGIC 68-SH 104-S 441-AARGAL 464-TPTA 508-D 510-C 512-VCT
4U9H: 34-E   81-CGVC 87-TH 123-D 477-APRGGL 500-VPST 544-D 546-C 548-ACG
```

**Extended Data Fig. 9 | Comparison of Huc with an O₂ sensitive [NiFe]-hydrogenase.** (a) A stereoview of the Huc [NiFe] cluster and surrounding amino acids. Amino acids conserved in O₂-sensitive hydrogenases and shown in blue, while divergent residues are shown in red. Amino acid type and position are labelled for divergent residues. (b) A stereoview overlay of the Huc catalytic site residues from panel a with those of the O₂-sensitive hydrogenase from *D. vulgaris* (PDB ID = 4U9H). Huc residues are coloured as in panel a, while *D. vulgaris* hydrogenase residues are coloured cyan (conserved) and pink (divergent). (c) A sequence alignment of Huc and *D. vulgaris* hydrogenase catalytic site residues, with divergent residues highlighted in red. (d) A cross-eye stereo view of D-histidine 166 from HucL shown as sticks demonstrating amino acid chirality. The corresponding Cryo-EM density map is shown at 5 σ, demonstrating the density is consistent with the D-isomer (top panel). L-histidine 194 form HucL is shown for reference (bottom panel). (e) The location of L-histidine 235 from the *D. vulgaris* hydrogenase, which corresponds to D-histidine 166 from Huc, showing its interaction with the proximal 4Fe-4S cluster. (f) An overlay of the relative positions of D-histidine 166 from Huc and L-histidine 235 from the *D. vulgaris* hydrogenase showing how the D-isomer shifts the position of the main chain in Huc.

**Extended Data Table 1 | Cryo-EM data collection, refinement and validation statistics**

| | #1 Huc Dimer (EMDB-26801) (PDB 7UUR) | #2 Huc Dimer (EMDB-27661) (PDB 8DQV) | #2 Huc Oligomer (EMDB-26767) (PDB 7UTD) | #2 Huc Tail (EMDB-26802) (PDB 7UUS) |
|---|---|---|---|---|
| **Data collection and processing** | | | | |
| Magnification | 105kx | 165kx | 105kx | 165kx |
| Voltage (kV) | 300 | 300 | 300 | 300 |
| Electron exposure (e–/Å$^2$) | 66.0 | 60.4 | 66.0 | 60.4 |
| Defocus range (μm) | -1.5-0.5 | -1.3-0.3 | -1.5-0.5 | -1.3-0.3 |
| Pixel size (Å) | 0.82 | 0.50 | 0.82 | 0.50 |
| Symmetry imposed | C2 | C2 | C4 | C1 |
| Initial particle images (no.) | 9,348,375 | 3,800,000 | 825,900 | 3,800,000 |
| Final particle images (no.) | 422,614 | 182,831 | 139,367 | 60,448 |
| Map resolution (Å) | 1.67 | 1.52 | 2.19 | 8.0 |
| FSC threshold | 0.143 | 0.143 | 0.143 | 0.143 |
| Map resolution range (Å) | 1.64-2.08 | 1.50-1.88 | 2.11-3.37 | 3.4-12.00 |
| | | | | |
| **Refinement** | | | | |
| Initial model used (PDB code) | - | - | - | - |
| Model resolution (Å) | 1.67 | 1.52 | 2.2 | n/a |
| FSC threshold | 0.143 | 0.143 | 0.143 | n/a |
| Model resolution range (Å) | n/a | n/a | n/a | n/a |
| Map sharpening $B$ factor (Å$^2$) | 36.1 | 25.7 | 57.2 | n/a |
| Model composition | | | | |
| Non-hydrogen atoms | 13,683 | 13,676 | 54,200 | 57,348 |
| Protein residues | 1,157 | 1,157 | 6,916 | 7,360 |
| Ligands | 12 | 12 | 48 | 48 |
| $B$ factors (Å$^2$) | | | | |
| Protein | 13.39 | 7.76 | 89.65 | 267.31 |
| Ligand | 11.29 | 6.99 | 64.72 | 220.27 |
| R.m.s. deviations | | | | |
| Bond lengths (Å) | 0.06 | 0.058 | 0.059 | 0.012 |
| Bond angles (°) | 2.336 | 2.289 | 2.45 | 1.96 |
| Validation | | | | |
| MolProbity score | 1.52 | 1.64 | 1.76 | 1.72 |
| Clashscore | 7.9 | 12.00 | 9.63 | 8.41 |
| Poor rotamers (%) | 0.58 | 0.22 | 1.69 | 1.67 |
| Ramachandran plot | | | | |
| Favored (%) | 97.59 | 97.83 | 97.62 | 97.53 |
| Allowed (%) | 2.17 | 1.99 | 2.15 | 2.29 |
| Disallowed (%) | 0.24 | 0.18 | 0.23 | 0.19 |

All datasets were collected with a zero-loss filtering slit width of 10 eV and with 60 frames per movie.

# Reporting Summary

## Statistics

For all statistical analyses, confirm that the following items are present in the figure legend, table legend, main text, or Methods section.

| n/a | Confirmed | |
|---|---|---|
| ☐ | ☒ | The exact sample size ($n$) for each experimental group/condition, given as a discrete number and unit of measurement |
| ☐ | ☒ | A statement on whether measurements were taken from distinct samples or whether the same sample was measured repeatedly |
| ☒ | ☐ | The statistical test(s) used AND whether they are one- or two-sided<br>*Only common tests should be described solely by name; describe more complex techniques in the Methods section.* |
| ☒ | ☐ | A description of all covariates tested |
| ☒ | ☐ | A description of any assumptions or corrections, such as tests of normality and adjustment for multiple comparisons |
| ☐ | ☒ | A full description of the statistical parameters including central tendency (e.g. means) or other basic estimates (e.g. regression coefficient) AND variation (e.g. standard deviation) or associated estimates of uncertainty (e.g. confidence intervals) |
| ☒ | ☐ | For null hypothesis testing, the test statistic (e.g. $F$, $t$, $r$) with confidence intervals, effect sizes, degrees of freedom and $P$ value noted<br>*Give P values as exact values whenever suitable.* |
| ☒ | ☐ | For Bayesian analysis, information on the choice of priors and Markov chain Monte Carlo settings |
| ☒ | ☐ | For hierarchical and complex designs, identification of the appropriate level for tests and full reporting of outcomes |
| ☒ | ☐ | Estimates of effect sizes (e.g. Cohen's $d$, Pearson's $r$), indicating how they were calculated |

*Our web collection on statistics for biologists contains articles on many of the points above.*

## Software and code

Policy information about availability of computer code

| | |
|---|---|
| Data collection | SensorTrace Suite v3.4.000, NOVA Autolab 1.10.0, GPES 4.9 |
| Data analysis | Microsoft Excel 2016, Graphpad Prism 8, Byonic (ProteinMetrics v4.3), UCSF Motioncor 1.0.4, CTFFIND 4.1.8, Relion 3.1.2, crYOLO 1.7.6 , cryoSPARC 3.0.1 and 3.3.2, gautomatch V 0.53, ChimeraX 1.3, Phenix 1.17, Alphafold 2.1.1, GROMACS v 2021.3, Origin 8, Qsoas 3.0, Coot 0.92, Pymol 2.4.1, pyem v0.5, CAVER 3.0, EasySpin v.6.0.0-dev47, |

For manuscripts utilizing custom algorithms or software that are central to the research but not yet described in published literature, software must be made available to editors and reviewers. We strongly encourage code deposition in a community repository (e.g. GitHub). See the Nature Portfolio guidelines for submitting code & software for further information.

## Data

Policy information about availability of data

All manuscripts must include a data availability statement. This statement should provide the following information, where applicable:
- Accession codes, unique identifiers, or web links for publicly available datasets
- A description of any restrictions on data availability
- For clinical datasets or third party data, please ensure that the statement adheres to our policy

Cryo-EM maps and atomic models generated in this paper have been deposited in the Electron Microscopy Data Bank (accession codes 7UTD, 7UUR, 7UUS, 8DQV) and the Protein Data Bank (accession codes EMD-26767, EMD-26801, EMD-26802, EMD-27661). Raw data from the Huc molecular dynamics simulations is available

via Zenodo (https://zenodo.org/record/7378976). Raw data numerical data for Huc activity, electrochemistry, and spectroscopy experiments generated during the current study are associated with the publication. Public available structural coordinates utilised in this study can be accessed via the following PDB accession codes, 5AA5, 2FRV, 4U9H, 6EHQ, 4C3O, 3AYX, 5MDK.

## Human research participants

Policy information about studies involving human research participants and Sex and Gender in Research.

| | |
|---|---|
| Reporting on sex and gender | not applicable |
| Population characteristics | not applicable |
| Recruitment | not applicable |
| Ethics oversight | not applicable |

Note that full information on the approval of the study protocol must also be provided in the manuscript.

## Field-specific reporting

Please select the one below that is the best fit for your research. If you are not sure, read the appropriate sections before making your selection.

☒ Life sciences          ☐ Behavioural & social sciences          ☐ Ecological, evolutionary & environmental sciences

For a reference copy of the document with all sections, see nature.com/documents/nr-reporting-summary-flat.pdf

## Life sciences study design

All studies must disclose on these points even when the disclosure is negative.

| | |
|---|---|
| Sample size | Sample size utilized and justification for why this is appropriate is provided in the 'Statistics and Reproducibility' section of the methods associated with this paper. |
| Data exclusions | A series of optimization experiments were conducted for enzyme assays, prior to the final generation of the final data presented in the manuscript. The data from these optimization experiments were not included in the final manuscript, however, they are consistent with the presented data. |
| Replication | The number of times experiments were replicated in this study is provided in the 'Statistics and Reproducibility' section of the methods associated with this paper. Where representative data is presented, all other replicates produced data that was consistent with the data presented. |
| Randomization | In this study we characterized a single purified enzyme (Huc hydrogenase) using a range of techniques. As a single enzyme sample was analysed randomization is not applicable to this study |
| Blinding | In this study we characterized a single purified enzyme (Huc hydrogenase) using a range of techniques. As a single enzyme sample was analysed blinding is not applicable to this study |

## Reporting for specific materials, systems and methods

We require information from authors about some types of materials, experimental systems and methods used in many studies. Here, indicate whether each material, system or method listed is relevant to your study. If you are not sure if a list item applies to your research, read the appropriate section before selecting a response.

### Materials & experimental systems

| n/a | Involved in the study |
|---|---|
| ☒ | ☐ Antibodies |
| ☒ | ☐ Eukaryotic cell lines |
| ☒ | ☐ Palaeontology and archaeology |
| ☒ | ☐ Animals and other organisms |
| ☒ | ☐ Clinical data |
| ☒ | ☐ Dual use research of concern |

### Methods

| n/a | Involved in the study |
|---|---|
| ☒ | ☐ ChIP-seq |
| ☒ | ☐ Flow cytometry |
| ☒ | ☐ MRI-based neuroimaging |

