## [Peer Review File · Nature]

Manuscript Title: Structural basis for bacterial energy extraction from atmospheric hydrogen

Reviewer Comments & Author Rebuttals

Reviewer Reports on the Initial Version:

Referees' comments:

Referee #1 (Remarks to the Author):

--It has been known for some time that many aerobic bacteria are able to oxidise atmospheric hydrogen by a [NiFe] hydrogenase, but how this feat is accomplished in the presence of the large amounts of the catalytic poison, oxygen, was unknown. Grinter et al. have now purified one of these enzymes, the mycobacterial hydrogenase Huc that couples hydrogen oxidation to menaquinone reduction. The authors showed that Huc has a high affinity for hydrogen and is insensitive to oxygen, and they determined its structure by cryo-EM to very high resolution. This made it possible to determine the structural basis for hydrogen delivery and oxygen insensitivity of the enzyme, which were verified by molecular dynamics simulations. Huc forms a large complex with 8 copies each of the large and small subunit, bound to a tetramer of a third subunit that forms a tubular membrane anchor. The tetramer forms a hydrophobic chamber for delivering the respiratory chain electron carrier menaquinone to the hydrogenase almost 100 Å above the membrane.

This paper shows a fascinating new structure and a convincing interpretation of the enzyme's mechanism. The work is technically sound, with one of the highest resolution cryo-EM structures to date. The paper is very well written and the data is supported by nice figures. It should find a broad audience among Nature readers.

The following points need attention:

1. Figure 2 showing the cryo-EM maps and the structural model is a bit confusing for various reasons. It is not obvious that the structure contains 8 copies each of HucS and HucL, and the membrane attachment mode is not clear. In detail:

a. The panels are in different orientations. 2c is 90 degrees rotated relative to 2a next to it as well as to 2b above it, while 2e next to 2c is turned by another 180 degrees. 2d and the right panel of 2f, showing the same view, are also 90 degrees rotated relative to each other. I suggest to be consistent and keep everything in a vertical orientation as in 2d, which means turning 2c, 2e and 2f.

b. Each of the four lobes consists of two HucSL dimers, but this is not clear from the figures. Due to the large size of HucL, one of the copies of HucS is obscured in both views in 2a. Only in the model in 2c both copies of HucS can be seen, but as they are shown in the same color it is not obvious. I suggest turning 2c 90 degrees counterclockwise and coloring the two copies of the subunits in different shades.

c. The representation of the composite map in figure 2d erroneously suggests that HucM is an integral membrane protein. As shown in figure S7, the bottom of the stalk contains positively charged residues interacting with the lipid headgroups, and the blob in the membrane results in fact from membrane vesicles (and is not as structured as it looks in figure 2d). It would be useful to show

the full model with the position of the membrane somewhere in the main figures, for example in the right panel of 2f (rotated to the same orientation of 2d). A good option would also be to color the stalk in 2d after fitting the AlphaFold model. It could even be in the four colors of 2e. This would make perfect sense, as the part of HucM visible in the “body” map is also colored in this figure. The lipid region could be grey (with preferably then also the Membrane box in grey).

d. In 2c “Ser70” is probably Ser20 (as in 2b). “Hucs” should be “HucS”.

e. The legends are incomplete. For example it is not mentioned that in 2a and 2d the blue/green lobe shows the high-resolution map as opposed to the full map. The NiFe center should be pointed out in 2c and/or 2f, as well as the significance of the indicated residues.

2. Figure 3a: the distances are not readable, really too small and pixelated. Same with the squiggles (text?) between the FeS clusters in fig. 4g, and the first set of FSC curves in fig. S4.

3. Unlike X-ray crystallography, cryo-EM does not produce electron density maps, but images the shielded Coulomb potential. Please replace “electron density” by “cryo-EM density” or just “density”, throughout the manuscript.

4. In the Methods there are two sections called “HucM tail reconstruction”, starting at line 844 and 855. Please check.

Minor points:

5. Line 188: two times “composed of” in one sentence. Please rewrite.

6. Fig. 4d legend: electron static should be electrostatic.

7. Fig. 4f legend: at menaquinone molecule should be a menaquinone molecule.

8. Line 383: “...become inhibited BY O2” (word missing).

9. Line 897: “The 1.52 Å crystallographic structure of Huc hydrogenase was used to construct the simulation system...”: I suppose the 1.52 Å cryo-EM structure is meant?

10. Figure S4-S6 and maybe elsewhere: homogenous refinement should be homogeneous refinement and heterogenous heterogeneous.

11. Fig. S5, text on right side: missing comma after box 576 pixels.

12. Fig. S7 legend: c-terminal should be C-terminal.

13. Fig. S8a: *Desulfovibrio Vulgaris* should be *Desulfovibrio vulgaris* (remove u, V lowercase).

Referee #2 (Remarks to the Author):

The authors reported biochemical and structural analysis of a new type of membrane-associating NiFe hydrogenase Huc, which oxidizes atmospheric concentration of H₂. The cryoEM structure indicated some interesting structural features of this enzyme, which correlate to the ability of oxidation of very low concentration of H₂: (i) the presence of 3 x [3Fe-4S]-cluster chain, which differ from standard NiFe-hydrogenases containing 3 x [4Fe-4S]-cluster chain, (ii) binding of menaquinone in the interior surface of the internal cavity of Huc, which should work as the high potential electron acceptor and (iii) the presence of D-form of histidine residue. These findings are highly interesting; however, to support the proposal, further experiments are required.

Specific comments:

1. The paragraph for O₂-insensitivity (from L. 208): The oxygen tolerance mechanism of this hydrogenase was extensively discussed in this section, which is based on the width of the substrate channel. However, this hypothesis has already been proposed previously for the other NiFe-hydrogenases as described by the authors in the text. Thus, most of the discussion of oxygen-tolerance is not new. This part should be substantially shortened.
2. Figure 1a: In this paper, the authors indicated that menaquinone is the physiological electron acceptor of Huc hydrogenase. In this context, the threshold of H₂ should be tested in the presence of menaquinone or menadione although the authors used NBT as described in the method section. In addition, to show that the H₂ threshold value of Huc is lower than the other NiFe hydrogenase, the data of standard NiFe hydrogenase should be tested in the same condition and compared in the figure.
3. Figure 1bc: The unit of the specific activity should be $\mu\text{mol}\cdot\text{min}^{-1}\cdot\text{mg}^{-1}$ rather than $\mu\text{M}\cdot\text{min}^{-1}\cdot\text{mg}^{-1}$.
4. Figure 1c: At L. 115, the authors wrote "Nevertheless, Huc does transfer electrons directly to O₂ (Fig. 1c)." What is the evidence? Figure 1c shows just enzyme kinetics using different electron acceptors.
5. Figure 1d. To show the distinct cyclic voltammogram of an immobilized Huc protein film, the same experiments of the standard NiFe-hydrogenase under the exactly same conditions should be shown.
6. L. 160. The Huc complex was mainly purified from soluble fraction in this study. However, a previous papers describes that the Huc complex is a membrane associated (Ref 5, 32). What is the reason of the difference of fractionation? How many percent of Huc was obtained in the membrane and soluble fractions in this work?
7. L. 174. Finding of three [3Fe-4S]-clusters are highly interesting. However, only the evidence of the presence of these clusters are the cryoEM map. I cannot judge the adequacy of this model from such a small figure. The authors should show larger density map and the model of [3Fe-4S]-clusters in supplementary materials.

8. L. 174. To prove the presence of such unique 3 x [3Fe-4S]-clusters, EPR spectroscopy data should be shown. To discuss the redox potential of the 3 x [3Fe-4S]-clusters, redox titration of the iron-sulfur clusters using EPR is required.
9. L. 174. The authors should mention the presence/absence of the fourth ligand residues connecting 3 x [4Fe-4S]-clusters in the other NiFe-hydrogenases, which indirectly indicate the absence of the [4Fe-4S]-clusters in Huc.
10. Figure S9. I guess that Ni-S (in the main text and Figure S9 legend) and Ni-SI (in Figure S9 panels) are the same state. Please unify.
11. Figure S9e. This result show that Huc in Ni-SI state converted to the Ni-B state by binding of oxygen and inhibited within ~400 s in the presence of 20% O₂. This result indicates that Huc is O₂ sensitive although throughout this manuscript, the authors call Huc as O₂-insensitive hydrogenase. Indeed, in short period, Huc hydrogenase shows O₂ insensitive property (Figure 1b); however, the name of O₂-insensitive hydrogenase is misleading, isn't it?
12. Figure S9G. The second molecule from top should be Ni-SI (or Ni-S) rather than Ni-C.
13. L. 294. Ref 66 concluded, "We find that "real" D-residues are rare among longer polypeptide chains (20 residues or more) formed from mostly L-amino acids in the PDB and, with a single exception, are the deliberate result of molecular design and protein engineering experiments. Thus, apparent D-residues in proteins are almost always either designed or are artifacts of the X-ray or NMR refinement process." If H166 is really the D-form, it is very interesting. However, to determine the D-form of histidine by cryo-EM structure analysis is difficult even with 1.52 Å resolution data. Much higher resolution structure analysis or different technique is required to distinguish the D/L forms.
14. Figure S10d. The structure of D- and L-histidine in Figure S10C are too small to see the structure and the density if the authors want to discuss the presence of the D-configuration.
15. Figure S10d. 1.5 Å resolution data does not appear to be enough to determine the H atom.
16. Figure 4a. The density maps of possible menaquinone are too small and too low-resolution to judge the adequacy of this model. You need to show larger figure with better resolution figure.
17. L. 316. The authors analyzed the purified enzyme with qualitative analysis of mass spectrometry and detected "high levels" of beta-dihydromenaquinone-9. To avoid the possibility of the unspecifically bound menaquinone, the authors should show the quantitative analysis (e.g. quantitative mass spectrometric analysis) of menaquinone in the sample and compare it with the concentration of the protein molecule in the sample to calculate the occupancy.
18. Figure S1d and its legend. The two panels of Figure S1d are the data of differential scanning fluorimetry (top) and differential scanning calorimetry (bottom). There is no description of the

method of differential scanning calorimetry in the method section. The legend of Figure S1d is incorrect.

Referee #3 (Remarks to the Author):

The study by Grinter and coworkers describes the isolation, biochemical characterization, high resolution Cryo-EM structure and mechanistic investigation of the mycobacterial hydrogenase Huc, belonging to an uncharacterized class of [NiFe] hydrogenases. This class is notably able to oxidize hydrogen at atmospheric (picomolar) concentrations in the presence of ambient concentrations of oxygen. This is in stark contrast to previously studied hydrogenases which are generally very oxygen sensitive, or at best, tolerant to some levels of oxygen. This is a discovery of great scientific interest with potential for applied science.

The authors propose a fundamentally different mechanism for the oxygen insensitivity by steric exclusion of the larger gas molecule. Oxygen tolerant hydrogenases rely on electrons from H₂ oxidation to reduce O₂ to water, a mechanism that would likely be impossible given the enormous concentration differences of the substrates in the current case.

There are three main parts to this study:

1. The isolation and in vitro biochemical characterization of the enzyme responsible for this activity, showing that the catalytic properties arise from the protein per se and are not dependent on additional systems or situations in the cell.
2. The remarkable Cryo-EM structure of the enzyme showing the cofactor arrangement as well as the overall functional architecture including potential (and surprising) binding mode of the electron acceptor menaquinone.
3. Structural and computational analysis and discussion of the enzymatic mechanism, in particular the key question of gas substrate access and a potential mechanism of exclusion of O₂ from the active site.

The methodology is standard and sound and the data quality and refinement statistics in the structural analysis appears solid and the Cryo-EM resolution achieved is remarkable. Structural, mass-spec and biochemical assay data are provided to support the conclusions.

The text is clear and easy to follow, though I feel it could benefit from shortening and polishing in places. The figures are clear and accessible, though figure 4 is very dense.

I would like to congratulate the authors to a very interesting discovery and a nice and complete study.

I would like to point out that while the computational studies provide a plausible explanation for the O₂ exclusion mechanism they cannot be viewed as conclusive. Mutational studies could be employed to prove the authors most central point and prediction regarding the mechanism for H₂ access and O₂ exclusion by removing bottlenecks in the gas access channels. Given the central importance of this issue such an experiment would strengthen the study. I do recognize, however that as the protein is chromosomally expressed these experiments would be more challenging than if expressed from a plasmid.

Other issues:

1. Throughout the paper the authors refer to “electron density map”. Maps derived by electron microscopy display Coulomb potential and not electron density, these are principally different and should be named correctly.
2. References 7 and 31 are either incomplete (7) or preprints (31) please correct and update with final reference if available.
3. Row 115 states “Huc does transfer electrons directly to O₂ (Fig. 1c).” I understand that this is most likely the case but i see no direct experiment or evidence for this.
4. I find the text rows 259-303 rather long and speculative, this would benefit from sharpening and shortening.
5. I find the mechanism for menaquinone access appealing but energetically problematic. Based on the figures it appears that a MQ would only interact with one side of the tunnel wall. Why would the MQ leave the fully hydrophobic membrane to access this channel which is presumably filled with solvent. An electron transfer event between bound redox sites (e.g quinones) would appear a more likely scenario to move electrons from the bound MQ to the membrane MQ pool. I feel figure 4g is too explicit for an unproven mechanism.

Author Rebuttals to Initial Comments:

Referees' comments:

Referee #1 (Remarks to the Author):

It has been known for some time that many aerobic bacteria are able to oxidize atmospheric hydrogen by a [NiFe] hydrogenase, but how this feat is accomplished in the presence of the large amounts of the catalytic poison, oxygen, was unknown. Grinter et al. have now purified one of these enzymes, the mycobacterial hydrogenase Huc that couples hydrogen oxidation to menaquinone reduction. The authors showed that Huc has a high affinity for hydrogen and is insensitive to oxygen, and they determined its structure by cryo-EM to very high resolution. This made it possible to determine the structural basis for hydrogen delivery and oxygen insensitivity of the enzyme, which were verified by molecular dynamics simulations. Huc forms a large complex with 8 copies each of the large and small subunit, bound to a tetramer of a third subunit that forms a tubular membrane anchor. The tetramer forms a hydrophobic chamber for delivering the respiratory chain electron carrier menaquinone to the hydrogenase almost 100 Å above the membrane.

This paper shows a fascinating new structure and a convincing interpretation of the enzyme's mechanism. The work is technically sound, with one of the highest resolution cryo-EM structures to date. The paper is very well written and the data is supported by nice figures. It should find a broad audience among Nature readers.

We thank the referee for the time spent reviewing our manuscript, their positive assessment of our work and helpful suggestions and insights to improve our manuscript.

The following points need attention:

1. Figure 2 showing the cryo-EM maps and the structural model is a bit confusing for various reasons. It is not obvious that the structure contains 8 copies each of HucS and HucL, and the membrane attachment mode is not clear. In detail:

a. The panels are in different orientations. 2c is 90 degrees rotated relative to 2a next to it as well as to 2b above it, while 2e next to 2c is turned by another 180 degrees. 2d and the right panel of 2f, showing the same view, are also 90 degrees rotated relative to each other. I suggest to be consistent and keep everything in a vertical orientation as in 2d, which means turning 2c, 2e and 2f.

We thank the reviewer for their insightful suggestions for improving this figure. We have rotated panels 2c, 2e and 2f in the revised figure, so that their orientation is consistent.

b. Each of the four lobes consists of two HucSL dimers, but this is not clear from the figures. Due to the large size of HucL, one of the copies of HucS is obscured in both views in 2a. Only in the model in 2c both copies of HucS can be seen, but as they are shown in the same color it is not obvious. I suggest turning 2c 90 degrees counterclockwise and coloring the two copies of the subunits in different shades.

In the revised figure we have applied different coloring to each of the HucSL dimers in panels 2a, 2c, 2d, and 2f. We have also improved the labelling of the subunits in these panels. We thank the reviewer for this suggestion and agree that this makes the figure much more interpretable.

c. The representation of the composite map in figure 2d erroneously suggests that HucM is an integral membrane protein. As shown in figure S7, the bottom of the stalk contains positively charged residues interacting with the lipid headgroups, and the blob in the membrane results in fact from membrane vesicles (and is not as structured as it looks in figure 2d). It would be useful to show the full model with the position of the membrane somewhere in the main figures, for example in the right panel of 2f (rotated to the same orientation of 2d). A good option would also be to color the stalk in 2d after fitting the AlphaFold model. It could even be in the four colors of 2e. This would make perfect sense, as the part of HucM visible in the “body” map is also colored in this figure. The lipid region could be grey (with preferably then also the Membrane box in grey).

We thank the reviewer for pointing out this ambiguity in figure 2. In the revised figure, we have followed the reviewer’s suggestions. In 2d we have coloured and labelled the HucM stalk region of the map to differentiate it from the membrane vesicle region of the map. We have added a box to 2f indicating the position of the membrane relative to the HucM stalk.

d. In 2c “Ser70” is probably Ser20 (as in 2b). “Hucs” should be “HucS”.

The reviewer is correct. We have corrected these errors in the revised figure.

e. The legends are incomplete. For example is not mentioned that in 2a and 2d the blue/green lobe shows the high-resolution map as opposed to the full map. The NiFe center should be pointed out in 2c and/or 2f, as well as the significance of the indicated residues.

These points have been addressed in the revised figure 2 and the figure legend (Lines 233-234)

2. Figure 3a: the distances are not readable, really too small and pixelated. Same with the squiggles (text?) between the FeS clusters in fig. 4g, and the first set of FSC curves in fig. S4.

The distances in figure 3a have been removed from the revised figure as they are too small to be rendered clearly. The size of the text in the revised figure 4i (previously 4g) has been increased. The resolution of figure S4 has been increased, so the FSC curve and now readable.

3. Unlike X-ray crystallography, cryo-EM does not produce electron density maps, but images the shielded Coulomb potential. Please replace “electron density” by “cryo-EM density” or just “density”, throughout the manuscript.

We thank the reviewer for pointing this out. We have replaced ‘electron density’ with ‘cryo-EM density’ or ‘density’ throughout the revised manuscript.

4. In the Methods there are two sections called “HucM tail reconstruction”, starting at line 844 and 855. Please check.

We attempted multiple approaches to reconstruct the HucM stalk region. In the initial manuscript, methods describing two of these approaches were accidentally included. We have removed one of these in the revised manuscript, only including the method that produced the maps we submitted to the EMDB and used for making figures for the paper. The method is shown on lines 1031-1049 in the revised manuscript.

Minor points:

5. Line 188: two times “composed of” in one sentence. Please rewrite.

Changed the first ‘composed of’ to ‘consisting of’ (Line 195)

6. Fig. 4d legend: electron static should be electrostatic.

Corrected (Line 366)

7. Fig. 4f legend: at menaquinone molecule should be a menaquinone molecule.

Corrected (Line 369)

8. Line 383: “...become inhibited BY O2” (word missing).

Corrected (Line 412)

9. Line 897: “The 1.52 Å crystallographic structure of Huc hydrogenase was used to construct the simulation system...”: I suppose the 1.52 Å cryo-EM structure is meant?

Yes, we did mean cryo-EM structure. This has been corrected in the revised manuscript (Line 1074)

10. Figure S4-S6 and maybe elsewhere: homogenous refinement should be homogeneous refinement and heterogenous heterogeneous.

This has been corrected in these figures (now figures S5, S6, and S8) and throughout the revised manuscript.

11. Fig. S5, text on right side: missing comma after box 576 pixels.

Corrected, now figure S8 of the revised manuscript

12. Fig. S7 legend: c-terminal should be C-terminal.

Corrected, now figure S9 legend of the revised manuscript (Line 553)

13. Fig. S8a: *Desulfovibrio Vulgaris* should be *Desulfovibrio vulgaris* (remove u, V lowercase).

Corrected, now figure S10 of the revised manuscript

Referee #2 (Remarks to the Author):

The authors reported biochemical and structural analysis of a new type of membrane-associated NiFe hydrogenase Huc, which oxidizes atmospheric concentration of H₂. The cryoEM structure indicated some interesting structural features of this enzyme, which correlate to the ability of oxidation of very low concentration of H₂: (i) the presence of 3 x [3Fe-4S]-cluster chain, which differ from standard NiFe-hydrogenases containing 3 x [4Fe-4S]-cluster chain, (ii) binding of menaquinone in the interior surface of the internal cavity of Huc, which should work as the high potential electron acceptor and (iii) the presence of D-form of histidine residue. These findings are highly interesting; however, to support the proposal, further experiments are required.

We thank the referee for their rigorous and insightful analysis of our manuscript and their suggestions for strengthening our study.

Specific comments:

1. The paragraph for O₂-insensitivity (from L. 208): The oxygen tolerance mechanism of this hydrogenase was extensively discussed in this section, which is based on the width of the substrate channel. However, this hypothesis has already been proposed previously for the other NiFe-hydrogenases as described by the authors in the text. Thus, most of the discussion of oxygen-tolerance is not new. This part should be substantially shortened.

This paragraph has been rewritten and merged with the subsequent paragraph so that it is more concise and focused. We have moved background information on previously characterised oxygen-tolerant hydrogenases into Supplementary note 2.

2. Figure 1a: In this paper, the authors indicated that menaquinone is the physiological electron acceptor of Huc hydrogenase. In this context, the threshold of H₂ should be tested in the presence of menaquinone or menadione, although the authors used NBT as described in the method section.

We have repeated the gas chromatography experiments with menadione as the electron acceptor and shown that Huc is able to oxidize H₂ to the limit of detection for our instrument (<0.04 ppmv) with this compound as the electron acceptor. It would be ideal to use menaquinone as the electron acceptor, but its highly hydrophobic nature means it is too poorly soluble to be used in this experiment. These data are shown in the revised figure 1a, and the gas chromatography analysis methods modified (Lines 927-945) to reflect these changes. In the initial manuscript, we opted to only present the data with NBT as an electron acceptor as menadione causes a decline in Huc activity after long (>12 hours) periods of incubation. We hypothesize that this is due small hydrophobic nature of menadione, which may cause it to act as a chaotrope. To overcome this and demonstrate subatmospheric H₂ oxidation with this electron acceptor, we increased the Huc concentration used in the assay

to 100 nM and reduced the starting H₂ concentration to 3 ppm. Under these conditions, Huc performed better than in our previous experiments, oxidizing H₂ to below the limited of detection for our gas chromatograph.

In addition, to show that the H₂ threshold value of Huc is lower than the other NiFe hydrogenase, the data of standard NiFe hydrogenase should be tested in the same condition and compared in the figure.

We thank the reviewer for this excellent suggestion. We have repeated the gas chromatography experiments, testing Huc activity in parallel with the oxygen-tolerant E. coli hydrogenase Hyd1. We concluded that Hyd1 was the best available ‘standard’ [NiFe]-hydrogenase to use for this experiment as it displays a reasonable degree of resistance to O₂. However, as Hyd1 is still inhibited by O₂ to some degree, so we conducted these experiments on anaerobically activated Hyd-1 in the absence of oxygen with a starting H₂ concentration of 10 or 100 ppm. While Huc was able to oxidize H₂ to near the limit of detection of our instrument (0.04 ppmv), Hyd1 was only able to oxidize H₂ to ~5 ppm. This comparison illustrates the high affinity of Huc compared to other [NiFe]-hydrogenases and strengthens the claims in our manuscript. These data have been included in Figure 1b and Figure S1f of the revised manuscript and discussed in lines 109-118.

3. Figure 1bc: The unit of the specific activity should be $\mu\text{mol}\cdot\text{min}^{-1}\cdot\text{mg}^{-1}$ rather than $\mu\text{M}\cdot\text{min}^{-1}\cdot\text{mg}^{-1}$.

We thank the reviewer for spotting this. It has been corrected in figure 1 of the revised manuscript.

4. Figure 1c: At L. 115, the authors wrote “Nevertheless, Huc does transfer electrons directly to O₂ (Fig. 1c).” What is the evidence? Figure 1c shows just enzyme kinetics using different electron acceptors.

Figure 1d in the revised manuscript shows the rate of Huc-mediated H₂ oxidation in N₂ purged buffer (No Acceptor, residual O₂ present) and O₂ saturated buffer (no additional electron acceptor, 100% O₂ saturation). These data show that O₂ stimulates Huc-mediated H₂ oxidation in the absence of another electron acceptor, which indicates that O₂ is acting as the electron acceptor. The rate of H₂ oxidation in O₂ saturated buffer is comparable to a buffer containing the non-specific electron acceptor NBT, and so we conclude that Huc is donating electrons from H₂ oxidation directly to O₂ at a rate comparable to this non-physiological electron acceptor.

5. Figure 1d. To show the distinct cyclic voltammogram of an immobilized Huc protein film, the same experiments of the standard NiFe-hydrogenase under the exactly same conditions should be shown.

We thank the reviewer for this excellent suggestion. We have repeated the protein film electrochemistry experiments, testing Huc in parallel with the E. coli hydrogenases Hyd1 and

Hyd2. These data clearly show the difference in onset potential and waveshape between Huc and the E. coli hydrogenases. In addition to comparing Huc to Hyd1 and Hyd2 we took this opportunity to assess the current response of Huc at H₂ concentrations closer to its physiological operating conditions, demonstrating that we can detect oxidising current from Huc as low as 10 ppm H₂. These data have been included in figure 1f and figure S2 of the revised manuscript and discussed on lines 166-185.

6. L. 160. The Huc complex was mainly purified from soluble fraction in this study. However, a previous paper describes that the Huc complex is a membrane associated (Ref 5, 32). What is the reason of the difference of fractionation? How many percent of Huc was obtained in the membrane and soluble fractions in this work?

Huc was solely purified from the soluble fraction in this work, with a portion of Huc molecules from this sample copurifying with small lipid vesicles. This is likely due to the formation of small Huc-associated lipid vesicles during the cell lysis step, which were not removed during centrifugation or size exclusion. The strain we utilized for Huc expression contains an inactivating mutant in the glycerol catabolism regulator GlyR. When this strain is grown on glycerol as its sole carbon source, it upregulates Huc expression for reasons that are currently unclear. This was advantageous for obtaining Huc for analysis, as it is not as abundantly expressed in the wild-type strain. We haven't precisely quantified the relative abundance of soluble vs membrane-bound Huc from these cells. However, from activity staining of membrane and soluble fractions, Huc appears to be largely associated with the soluble fraction (>95%). As the reviewer points out, this contrasts with Huc expressed in wild-type cells, which is largely membrane-associated via HucM. We are unsure why this is the case, but we suspect that the overexpression of Huc is what is driving its soluble localization. This may be due to Huc requiring an unidentified interaction partner to stably anchor it to the membrane surface, which may become saturated when Huc is overexpressed.

7. L. 174. Finding of three [3Fe-4S]-clusters are highly interesting. However, only the evidence of the presence of these clusters are the cryoEM map. I cannot judge the adequacy of this model from such a small figure. The authors should show larger density map and the model of [3Fe-4S]-clusters in supplementary materials.

We agree with the reviewer that the cryo-EM density map figures in the original manuscript were too small for assessing the modelling of the HucS [3Fe-4S] clusters. To clearly show the [3Fe-4S] cluster density, we have added larger cross-eye stereo panels showing the density of this region in figure S7 and an animation of this region in movies S2-4.

8. L. 174. To prove the presence of such unique 3 x [3Fe-4S]-clusters, EPR spectroscopy data should be shown.

We now performed EPR spectroscopy on air-oxidized, and H₂-reduced Huc. These data strongly support our assignment based on the structural data of all HucS clusters as having a [3Fe-4S] configuration and confirm they are redox-active. These data are presented in figure 3c and figure S7b,c,d and discussed in the revised manuscript on lines 207-214 revised manuscript and in Supplemental note 1 (Lines 652-679).

To discuss the redox potential of the 3 x [3Fe-4S]-clusters, redox titration of the iron-sulfur clusters using EPR is required.

We agree with the reviewer that an EPR derived redox titration of Huc would be informative for determining the potentials of the HucS [3Fe-4S] clusters. However, this experiment is very sample intensive, with approximately 15 mg of protein required for a full titration. As our average yield of Huc is approximately 20 ug per litre of culture this would require a purification from approximately 750 litres of culture to obtain the Huc required for this experiment, which isn't feasible given the resources we have available. In the current manuscript we don't discuss the specific redox potential of the HucS FeS clusters, and only make the general statement that they are likely to be a higher potential due to their [3Fe-4S] configuration, which is supported by the literature [1] and the overall redox properties of Huc. Therefore, we feel that these experiments are outside the scope of the paper.

9. L. 174. The authors should mention the presence/absence of the fourth ligand residues connecting 3 x [4Fe-4S]-clusters in the other NiFe-hydrogenases, which indirectly indicates the absence of the [4Fe-4S]-clusters in Huc.

The nature of the amino acids in these positions and their interaction with each of the [3Fe-4S] clusters in HucS is now shown in figure S7a and movies S2-4. We have added the following to the revised manuscript (lines 205-206) indicating the nature of these residues supports our assignment of the clusters:

Density maps and the identity of fourth coordinating residue of the three iron-sulfur clusters of the small subunit indicates they all have a [3Fe-4S] configuration (Fig. 3b, Fig. S7a, Mov S2,S3,S4).

10. Figure S9. I guess that Ni-S (in the main text and Figure S9 legend) and Ni-SI (in Figure S9 panels) are the same states. Please unify.

SI is the correct name for this state. This has been corrected in the revised manuscript, now figure S12, lines 591-595.

11. Figure S9e. This result show that Huc in Ni-SI state is converted to the Ni-B state by binding of oxygen and inhibited within ~400 s in the presence of 20% O₂. This result indicates that Huc is O₂ sensitive although throughout this manuscript, the authors call Huc as O₂-insensitive hydrogenase. Indeed, in short periods, Huc hydrogenase shows O₂ insensitive property (Figure 1b); however, the name of O₂-insensitive hydrogenase is misleading, isn't it?

Our activity assays show that Huc is active when exposed to O₂ at atmospheric concentrations (~)21% over an indefinite period, as the protein is purified, handled, stored, and activity assays are conducted in ambient air. Further, we do not observe any significant change in Huc H₂ oxidation kinetics in a buffer containing between ~0% to 100% O₂ saturation, which indicates there is no O₂ concentration dependent inhibition or change in Huc activity (figure 1c). The FTIR data indicates that Huc adopts a predominantly Ni-B state in the presence of 20% O₂. However, as this is approximately same concentration of O₂ that Huc is handled and assayed under, this indicates that for Huc, this Ni-B state is not inhibited in regard to H₂ oxidation. It is currently unclear how this is achieved, and it will be a

fascinating topic for future study.

12. Figure S9G. The second molecule from top should be Ni-SI (or Ni-S) rather than Ni-C.

Thank you. This has been corrected in the revised manuscript, now figure S12.

13. L. 294. Ref 66 concluded, “We find that “real” D-residues are rare among longer polypeptide chains (20 residues or more) formed from mostly L-amino acids in the PDB and, with a single exception, are the deliberate result of molecular design and protein engineering experiments. Thus, apparent D-residues in proteins are almost always either designed or are artifacts of the X-ray or NMR refinement process.” If H166 is really the D-form, it is very interesting. However, to determine the D-form of histidine by cryo-EM structure analysis is difficult even with 1.52 Å resolution data. Much higher resolution structure analysis or different technique is required to distinguish the D/L forms.

14. Figure S10d. The structure of D- and L-histidine in Figure S10C are too small to see the structure and the density if the authors want to discuss the presence of the D-configuration.

15. Figure S10d. 1.5 Å resolution data does not appear to be enough to determine the H atom.

We agree with the reviewer that the images we provided in the initial submission were inadequate for clearly showing the D-isomeric form of HucL His-166. In the revised submission, we have included improved cross-eye stereo images of this region and that of L-HucL His193 for reference in figure S13d, and an animation of the density from these regions in movies S6-7. Because of the different interactions of X-rays and electrons with matter, using cryo-EM, hydrogen atoms are visible at a much lower resolution than in X-ray crystallography. This is described in a recent Nature paper on high-resolution Cryo-EM by Yip et al. [2]:

“In X-ray crystallography, it is particularly challenging to see hydrogen atoms because of the limited photon scattering power of the single electron in a hydrogen atom. The situation in cryo-EM is different because electrons are scattered by the nuclear potential, which—in the case of H atoms—results in a larger scattering cross-section compared to photons in X-ray crystallography. Whereas H atoms become visible in X-ray maps typically only at resolutions close to 1 Å or better, they are expected to become visible at somewhat lower resolution in cryo-EM”

As there are relatively few high-resolution Cryo-EM structures, the resolution at which hydrogen atoms routinely become visible has not been rigorously established, and factors other than the stated map resolution contribute to their observation. However, as can be seen from the images and videos provided with our revised submission, the density of many hydrogen atoms can be observed in our structure.

With the improved visualization in the revised manuscript, the case for the presence of HucL D-His166 is compelling. It is impossible to fit the available electron density if His-166 is modelled as an L-isomer, and so we conclude that this case represents the largely unprecedented occurrence of a D-isomeric modification or substitution in a large protein.

16. Figure 4a. The density maps of possible menaquinone are too small and too low-resolution to judge the adequacy of this model. You need to show larger figure with better resolution figure.

We have revised figure 4 to increase the size of these panels. The low figure resolution is a result of the processing by the manuscript handling system, but high-resolution versions of the images have been submitted and should be downloadable. The final published images will be high resolution. Density for the menaquinone head group from the 1.52 Å reconstruction is also shown in movie S8.

17. L. 316. The authors analyzed the purified enzyme with qualitative analysis of mass spectrometry and detected “high levels” of beta-dihydromenaquinone-9. To avoid the possibility of the unspecifically bound menaquinone, the authors should show the quantitative analysis (e.g. quantitative mass spectrometric analysis) of menaquinone in the sample and compare it with the concentration of the protein molecule in the sample to calculate the occupancy.

We agree with the reviewer that it is important to quantitate the amount of beta-dihydromenaquinone-9 (MQ9(II-H₂)) present in purified Huc. We have now run commercially available menaquinone 9 (MQ9), which differs from (MQ9(II-H₂)) by a single additional double bond, as a standard at a concentration equivalent to the MQ9(II-H₂) that would be present in the Huc sample if it was 100% occupancy. Based on the area of both peaks, we calculate an MQ9(II-H₂) occupancy of 96.2% in the Huc sample. We have presented these data in Figure S14, and a summary of our calculations in the methods section (Lines 891-904).

18. Figure S1d and its legend. The two panels of Figure S1d are the data of differential scanning fluorimetry (top) and differential scanning calorimetry (bottom). There is no description of the method of differential scanning calorimetry in the method section. The legend of Figure S1d is incorrect.

We apologise for this error. We indeed performed differential scanning fluorimetry, not calorimetry. We have corrected in the revised legend for Figure S1 (Lines 472-473).

Referee #3 (Remarks to the Author):

The study by Grinter and coworkers describes the isolation, biochemical characterization, high resolution Cryo-EM structure and mechanistic investigation of the mycobacterial hydrogenase Huc, belonging to an uncharacterized class of [NiFe] hydrogenases. This class is notably able to oxidize hydrogen at atmospheric (picomolar) concentrations in the presence of ambient concentrations of oxygen. This is in stark contrast to previously studied hydrogenases which are generally very oxygen sensitive, or at best, tolerant to some levels of oxygen. This is a discovery of great scientific interest with potential for applied science. The authors propose a fundamentally different mechanism for the oxygen insensitivity by steric exclusion of the larger gas molecule. Oxygen tolerant hydrogenases rely on electrons from H₂ oxidation to reduce O₂ to water, a mechanism that would likely be impossible given the enormous concentration differences of the substrates in the current case.

There are three main parts to this study:

1. The isolation and in vitro biochemical characterization of the enzyme responsible for this activity, showing that the catalytic properties arise from the protein per se and are not dependent on additional systems or situations in the cell.
2. The remarkable Cryo-EM structure of the enzyme showing the cofactor arrangement as well as the overall functional architecture including potential (and surprising) binding mode of the electron acceptor menaquinone.
3. Structural and computational analysis and discussion of the enzymatic mechanism, in particular the key question of gas substrate access and a potential mechanism of exclusion of O₂ from the active site.

The methodology is standard and sound and the data quality and refinement statistics in the structural analysis appears solid and the Cryo-EM resolution achieved is remarkable. Structural, mass-spec and biochemical assay data are provided to support the conclusions. The text is clear and easy to follow, though I feel it could benefit from shortening and polishing in places. The figures are clear and accessible, though figure 4 is very dense.

I would like to congratulate the authors to a very interesting discovery and a nice and complete study.

We thank the referee for their thorough review of our manuscript, their positive comments about the study and their insightful suggestions for improving our work.

I would like to point out that while the computational studies provide a plausible explanation for the O₂ exclusion mechanism they cannot be viewed as conclusive. Mutational studies could be employed to prove the authors most central point and prediction regarding the mechanism for H₂ access and O₂ exclusion by removing bottlenecks in the gas access channels. Given the central importance of this issue such an experiment would strengthen the study. I do recognize, however that as the protein is chromosomally expressed these experiments would be more challenging than if expressed from a plasmid.

We agree with the reviewer that mutagenesis showing that the Huc gas channels restrict O₂ access to the Huc active site would strengthen this hypothesis, which is currently based on structural data and molecular dynamics simulations. In an attempt to produce these data, we generated three mutants that, based on analysis of the Huc structure, we predict would widen the bottleneck that excludes O₂ from entering the Huc active site. As the reviewer indicated, we needed to generate these mutants chromosomally, and so we elected to make 2 double (HucL E15A and I64A; HucL I64A and L112A), and 1 triple mutant (HucL E15A, I64A and L112A) as we suspected that a single mutant wouldn't sufficiently widen the gas channel to observe a change in the properties of Huc.

Unfortunately, despite the considerable effort in constructing these mutants, none of the resulting Huc variants were produced at detectable levels, either by activity staining of cell lysates (see Fig. R1 below) or through isolation from large-scale cultures via streptactin affinity chromatography. We suspect that these mutants destabilize HucL sufficiently to prevent folding, leading to its degradation. This result is indicative of the difficulty of using mutagenesis to analyze the function of residues in the tightly packed core of [NiFe]-hydrogenases.

While we were unable to support our hypothesis experimentally, we performed further molecular dynamics simulations on these Huc mutant variants, which showed that for two of the mutants (E15A and I64A; E15A, I64A and L112A), O₂ reaches the enzyme active site.

We have included these data in Figure S11 and revisions to the main text (Lines 269-270) supplemental note 3 (Lines 737-748).

Figure R1 NBT-based hydrogenase activity staining of cell lysates of *M. smegmatis* containing parent wild-type and derived gas channel mutant Huc proteins. Clarified cell lysate (containing 30 μ g of protein) from each strain was run on a 4-16% native-PAGE gel. The gel was subsequently placed in a 500 μ M NBT solution and incubated under a 7% H_2 atmosphere. Staining results from H_2 -mediated reduction of NBT by Huc. The lack of Huc activity in strains containing Huc mutants is a result of the protein no longer being produced in detectable quantities.

Other issues:

1. Throughout the paper the authors refer to “electron density map”. Maps derived by electron microscopy display Coulomb potential and not electron density, these are principally different and should be named correctly.

We thank the reviewer for pointing this out. ‘Electron density map’ has been changed to ‘Cryo-EM density map’ or ‘density map’ throughout the manuscript

2. References 7 and 31 are either incomplete (7) or preprints (31) please correct and update with final reference if available.

These references have been updated in the revised manuscript (Refs 7 and 31).

3. Row 115 states “Huc does transfer electrons directly to O_2 (Fig. 1c).” I understand that this is most likely the case but i see no direct experiment or evidence for this.

We have addressed this comment in response to reviewer 2 (point 4). O_2 strongly stimulates the rate of H_2 oxidation by Huc in the absence of another electron acceptor, with a response nearly identical to the use of NBT as an electron acceptor. We feel this is strong evidence of the direct transfer of H_2 -derived electrons from Huc to O_2 .

4. I find the text rows 259-303 rather long and speculative. This would benefit from sharpening and shortening.

We have shortened and rewritten this paragraph as suggested, including moving material to the new Supplemental note 2.

5. I find the mechanism for menaquinone access appealing but energetically problematic. Based on the figures it appears that a MQ would only interact with one side of the tunnel wall. Why would the MQ leave the fully hydrophobic membrane to access this channel which is presumably filled with solvent. An electron transfer event between bound redox sites (e.g quinones) would appear a more likely scenario to move electrons from the bound MQ to the membrane MQ pool. I feel figure 4g is too explicit for an unproven mechanism.

While further work needs to be done to validate our model for MQ access to the Huc active site, we feel that it represents the simplest and most plausible explanation for how this process functions. As shown in Figure S9c, the interior of the Huc tail contains additional density that likely corresponds to poorly ordered hydrophobic molecules and given the dimensions of the tube and the hydrophobicity of its internal surface, it's unlikely the tunnel is solvated. So there shouldn't be a significant energetic barrier to menaquinone entering the tunnel. The central chamber of Huc may contain solvent. However, there is density associated with the walls of the chamber that very likely corresponds to hydrophobic molecules, as the wall of the chamber is lined with hydrophobic residues. The opposite face of these hydrophobic molecules is, in turn, unlikely to be solvent exposed, and so possibly, the entire chamber may be filled with lipids, which aren't clearly resolved in the EM maps due to disorder. In support of this possibility, as shown in Figure 4g, the narrow region at the top of the Huc chamber contains density consistent with the aliphatic chains of lipid molecules.

Two additional points support our model:

Firstly, MQ is bound to all the electron acceptor sites of Huc, at high occupancy, as shown by the mass spectrometry quantification included in the revised manuscript, and it must have got there somehow. The simplest explanation would be that MQ reaches the active site via the hydrophobic tunnel, which indicates that this path to the acceptor site is viable for MQ access.

Secondly, for electron transfer to occur efficiently between redox-active molecules, they must be stably coordinated within the electron transfer distance. There is no density that indicates the presence of an electron transfer relay from the MQ at the electron acceptor site to the membrane, indicating this is unlikely the mechanism at play.

However, to acknowledge that this is our best working model of how Huc functions, we have modified the text to soften the certainty of our conclusions:

*Analysis of the structure of the Huc oligomer indicates **a possible** mechanism by which this occurs. (Line 375)*

*The conduit for menaquinone entry to and exit from the hydrophobic chamber is **likely** provided by the membrane-associated Huc stalk. (Lines 380-81)*

*..., which **could** enter through the interaction of HucM with the cell membrane (Mov. S10). (Line 383)*

*Reduced menaquinol then **could diffuse** back into the membrane via the HucM C-terminal tube,... (Lines 386-387)*

References

1. Zanello P. Structure and electrochemistry of proteins harboring iron-sulfur clusters of different nuclearities. Part II. [4Fe-4S] and [3Fe-4S] iron-sulfur proteins. *J Struct Biol.* 2018;202(3):250-63.
2. Yip KM, Fischer N, Paknia E, Chari A, Stark H. Atomic-resolution protein structure determination by cryo-EM. *Nature.* 2020;587(7832):157-61.

Reviewer Reports on the First Revision:

Referees' comments:

Referee #2 (Remarks to the Author):

I have reviewed the responses and the revised manuscript. I am happy with the answers provided by the authors.

Referee #3 (Remarks to the Author):

The authors have addressed my concerns.

I particularly appreciate the effort to produce mutant proteins for experimental verification of the mechanism. Regrettably these were not possible to express and purify, but in my opinion the authors have done what is reasonably possible in this case.

The added simulations also provide some further support.

I also appreciate the softening of the text regarding the mechanistic proposal for electron transfer.

One final point, As reviewer #2 I also find the identification of a D-amino acid in a protein truly remarkable. While I agree that the density displayed in movies S6 and S7 looks very convincing, significant conformational strain in proteins can result in surprising geometries and misinterpretations of density. Without further experimental evidence I would strongly suggest a more careful description of this observation in the text.

Author Rebuttals to First Revision:

Referees' comments:

Referee #2 (Remarks to the Author):

I have reviewed the responses and the revised manuscript. I am happy with the answers provided by the authors.

We thank the referee for again taking the time to review our manuscript. We are pleased that they are satisfied with the revisions we have made. Their helpful suggestions and comments have considerably strengthened the final submission.

Referee #3 (Remarks to the Author):

The authors have addressed my concerns.

I particularly appreciate the effort to produce mutant proteins for experimental verification of the mechanism. Regrettably these were not possible to express and purify, but in my opinion the authors have done what is reasonably possible in this case.

The added simulations also provide some further support.

I also appreciate the softening of the text regarding the mechanistic proposal for electron transfer.

The thank the referee for again reviewing our manuscript and are pleased they appreciate our efforts to incorporate their comments and suggestions.

One final point, As reviewer #2 i also find the identification of a D-amino acid in a protein truly remarkable. While I agree that the density displayed in movies S6 and S7 looks very convincing, significant conformational strain in proteins can result in surprising geometries and misinterpretations of density. Without further experimental evidence I would strongly suggest a more careful description of this observation in the text.

We agree that the D-histidine in Huc is remarkable. While we are confident our assignment of this stereo isomer is correct, we concede there is an outside chance that there could be another explanation for the observed density. As a result, we have softened our conclusions in the revised manuscript by making the following changes:

- *We have removed 'and their unusual ligation by a D-histidine' from the summary paragraph*
- *We have added the following bold text to the result section (lines 237-38): The redox properties of the Huc proximal and medial clusters may be further modified by an unusual D-isomer of histidine, **indicated by the cryo-EM maps**, at position 166 of HucL.*